# Geographics and bacterial networks differently shape the acquired and latent global sewage resistomes

Hannah-Marie Martiny [1,123] ✉, Patrick Munk [1,123] ✉, Alessandro Fuschi [2], Ágnes Becsei[3], Nikiforos Pyrounakis[1], Christian Brinch [1], Global Sewage Consortium*, D. G. Joakim Larsson [4], Marion Koopmans [5], Daniel Remondini [2], István Csabai[3] & Frank M. Aarestrup [1]

Antimicrobial resistance genes (ARGs) have rapidly emerged and spread globally, but the pathways driving their spread remain poorly understood. We analyzed 1240 sewage samples from 351 cities across 111 countries, comparing ARGs known to be mobilized with those identified through functional metagenomics (FG). FG ARGs showed stronger associations with bacterial taxa than the acquired ARGs. Network analyses further confirmed this and showed potential for source attribution of both known and novel ARGs. The FG resistome was more evenly dispersed globally, whereas the acquired resistome followed distinct geographical patterns. City-wise distance-decay analyses revealed that the FG ARGs showed significant decay within countries but not across regions or globally. In contrast, acquired ARGs showed decay at both national and regional scales. At the variant level, both ARG groups had significant national and regional distance-decay effects, but only FG ARGs at a global scale. Additionally, we observed stronger distance effects in Sub-Saharan Africa and East Asia compared to North America. Our findings suggest that differential selection and niche competition, rather than dispersal, shape the global resistome patterns. A limited number of bacterial taxa may act as reservoirs of latent FG ARGs, highlighting the need of targeted surveillance to mitigate future resistance threats.

Acquired resistance to antimicrobials among bacteria is a significant and increasing problem. It is difficult to assess the global burden of antimicrobial resistance (AMR), but global mortality has been estimated to be 1.14 million deaths in 2021[1], with African regions being disproportionately affected and estimated to increase to 1.91 M by 2050[2]. Another study predicts an average loss of 1.8 years of life expectancy globally by 2035, along with an additional cost of US$ 855 billion per year in extra healthcare costs and lost workforce productivity, or ~1% of the global economy[3].

Antimicrobial resistance genes (ARGs) are ancient and naturally occur in bacteria across diverse environments[4–9]; however, their rapid emergence in the clinic and global dissemination have been largely driven by human activities. Most resistome studies focus on those so-called acquired ARGs that have been mobilized from their origin and

---

[1]Research Group for Genomic Epidemiology, Technical University of Denmark, Lyngby, Denmark. [2]Department of Physics and Astronomy (DIFA), University of Bologna, Bologna, Italy. [3]Department of Physics of Complex Systems, ELTE Eötvös Loránd University, Budapest, Hungary. [4]Department of Infectious Diseases, Institute of Biomedicine, University of Gothenburg, and Centre for Antibiotic Resistance Research in Gothenburg, Gothenburg, Sweden. [5]Erasmus Medical Centre, Rotterdam, The Netherlands. [123]These authors contributed equally: Hannah-Marie Martiny, Patrick Munk.*A list of authors and their affiliations appears at the end of the paper. ✉e-mail: hanmar@food.dtu.dk; pmun@food.dtu.dk

transferred between species[10–12], but this likely only represents a small subset of the total resistome[10], where it has been suggested that only between 5 and 15% of all ARGs are being studied, as predicted in silico[11].

In silico prediction does not prove the functionality of a given gene. However, another method of discovering new ARGs is functional metagenomics (FG), which, based on random cloning and phenotypic selection, has revealed a diverse resistome across many bacterial communities[13,14]. While many of these novel ARGs, or FG ARGs, identified either in silico or in vitro, likely are intrinsic genes of environmental bacteria and have not been acquired by other bacteria, they represent a latent reservoir of resistance that might be mobilized in the future[11,13]. Several studies have shown that ARGs are subjected to dispersal limitations within and between reservoirs[15–17], but a systematic comparison of the distributions of acquired ARGs and FG ARGs has, to our knowledge, not yet been performed.

Epidemiological tracking of ARGs could help build a general understanding of how ARGs evolve, change hosts, and disseminate throughout our ecosphere, and inform interventions and policies[18,19]. Global compatible data, including bacterial genomic data, are, however, scarce. Sewage offers a convenient and ethical way of monitoring AMR, as it integrates waste from humans, their animals, and the surrounding environment[20]. We and others have recently utilized human sewage to monitor AMR in large, mainly healthy human populations[15,16,20]. These sewage resistome studies have found systematic differences in the abundance and diversity of acquired ARGs between the different world regions and much variation that can neither be explained by antimicrobial use nor bacterial taxonomic composition, but more strongly correlated with socioeconomic, health, and environmental factors[16,21,22]. Analysis of flanking sequences of a subset of acquired ARGs has suggested each ARG has unique patterns of dispersal limitation and global transmission, while other studies have identified mobilization differences as a function of resistance mechanism[23].

Dividing the world into regions typically represents a grouping of cultural, economic, historical, political, or other factors that can provide valuable insights into population differences but might be of little relevance concerning the dispersal limitations of ARGs. Thus, the question remains whether it is such regional factors or simply geographical distances that limit dispersal. Distance-decay relationships can reveal the effects of such spatial processes[24]. Studies have shown that human activities significantly influenced the presence of ARGs on coasts, but their similarity decreased as they dispersed into estuaries and the open sea, indicating strong distance-decay gradients along aquatic environments[24,25]. Most studies of distance-decay stick to a single region, where it can be difficult to investigate whether events on one side of the globe affect the other. However, a single study looked into the resistomes of globally distributed lake sediments and found that ARG compositions exhibited distance-decay relationships but were largely shaped by bacterial community structure[25].

As part of our recent efforts to combine multiple existing collections of ARG references into a single database called PanRes[26], we also included two collections of ARGs identified through functional cloning, namely ResFinderFG 2.0[14] and the collection from Daruka et al.[13]. We hypothesize that these ARGs identified through FG would be more strongly associated with environmental bacterial taxa and possibly show more substantial dispersal limitations than those ARGs that have mobilized, in relation to our human-associated bacteria, and could travel globally with us.

In this study, we aimed to systematically compare the abundance, diversity, and bacterial associations of acquired ARGs with those ARGs identified with FG. We expanded our collection of global sewage datasets to cover samples between 2016 and 2021. Using this dataset, we characterized the resistomes of 3131 acquired ARGs from ResFinder[19] and 4990 FG ARGs from ResFinderFG 2.0[14] and Daruka et al.[13]. While the results confirmed the regional clustering of acquired ARGs as shown in our previous studies[15,16], we found, contrary to our hypothesis, that the FG ARGs were much more evenly distributed across the globe. Only in Sub-Saharan Africa were the relative abundances of acquired and FG ARGs equal. Our distance-decay analyses revealed that while the FG ARGs exhibited a continuous decay, distance did not appear to affect dispersal at the inter-regional scale. The network analyses revealed that the FG ARGs were strongly linked with the underlying sewage bacteriomes, suggesting that the majority of the FG ARGs represent a latent reservoir of resistance. Together, these findings highlight the importance of monitoring both the acquired and FG ARGs to address the current and future threats of AMR.

## Results

### Summary of sewage sample metagenomes

The urban sewage collection included 1240 samples from 351 cities across 111 countries, spanning all seven world regions from 2016 to 2021 (Supplementary Fig. 1). On average, each sample had 32.39 million (M) trimmed sequence fragments (range: 0.03–515.56 (M), std: 21.82 M), totaling more than $8.93 \times 10^{10}$ read fragments. All trimmed fragments were mapped and aligned against the mOTUs conserved marker genes and the PanRes ARGs. 0.16% of these read fragments were assigned to mOTUs and 0.04% to PanRes ARGs. In our chosen subset of the PanRes collections, 0.019% trimmed read fragments were aligned to acquired ARGs, and 0.024% to FG ARGs. See the Supplementary Notes for the distribution of all PanRes ARGs.

A total of 141.07 M fragments were assigned to mOTUs (sample average: 0.10 M, std: 0.05 M), of these 99.4% were bacteria, 0.002% eukaryotes, and 0.56% archaea. For the bacterial hits, the fragments matched 13324 genera; however, 10,974 of them were placeholder names, and 2350 were latinized bacterial genera. The most common genera were *Pseudomonas* (2.91% of bacterial reads), *Acinetobacter* (2.44%), *Acidovorax* (1.77%), *Neisseria* (1.36%), and *Streptococcus* (0.87%).

We found evidence of 1052 acquired ARGs and 3095 FG ARGs, totaling 17.28 M fragments and 21.75 M fragments, respectively. On average, a sample had 0.015 M fragments to acquired ARGs (range: 0.000214–0.113 M, std: 0.013 M) and 0.019 M to FG ARGs (range: 0.000612–0.239 M, std: 0.016 M) (Supplementary Fig. 4a). 201 acquired ARGs and 527 FG ARGs were present in at least 50% of the samples with resistance fragments, with 1 acquired ARG (*aph(6)-id_2*) and 6 FG ARGs being present in all.

### Functionally identified ARGs are more abundant and geographically widespread

In agreement with our previous findings on a subset of the samples, the acquired ARGs were most abundant in Sub-Saharan Africa (SSA), the Middle East & North Africa (MENA), and South Asia (SA) (Fig. 1a). FG ARGs showed a higher and more evenly distributed abundance across the regions and countries than the acquired ARGs, with particularly high abundances in SSA and MENA (Fig. 1a, Supplementary Fig. 5, Supplementary Fig. 6). There were also a few cases where the acquired ARG load was high enough to approach FG loads, as seen in Cambodia and Iran (Supplementary Fig. 5). Alpha diversity analyses showed a closer relationship between the bacteriome and the FG resistomes compared to the acquired resistomes (Supplementary Fig. 4).

We estimated the regional pan- and core-resistomes. Overall, the sizes of the FG pan- and core-resistomes were larger than the acquired resistome (Supplementary Fig. 6), likely due to a combination of more FG references than acquired and an overall higher abundance (Supplementary Fig. 3, Supplementary Fig. 4). The core resistome constituted 12% and 23% of the pan-resistomes for FG and acquired genes, respectively.

Overall, the abundance and beta diversity of the acquired ARGs reflected the world regions, with 12% of the beta diversity explained by regional grouping (permanova, $p = 0.001$). In contrast, world regions

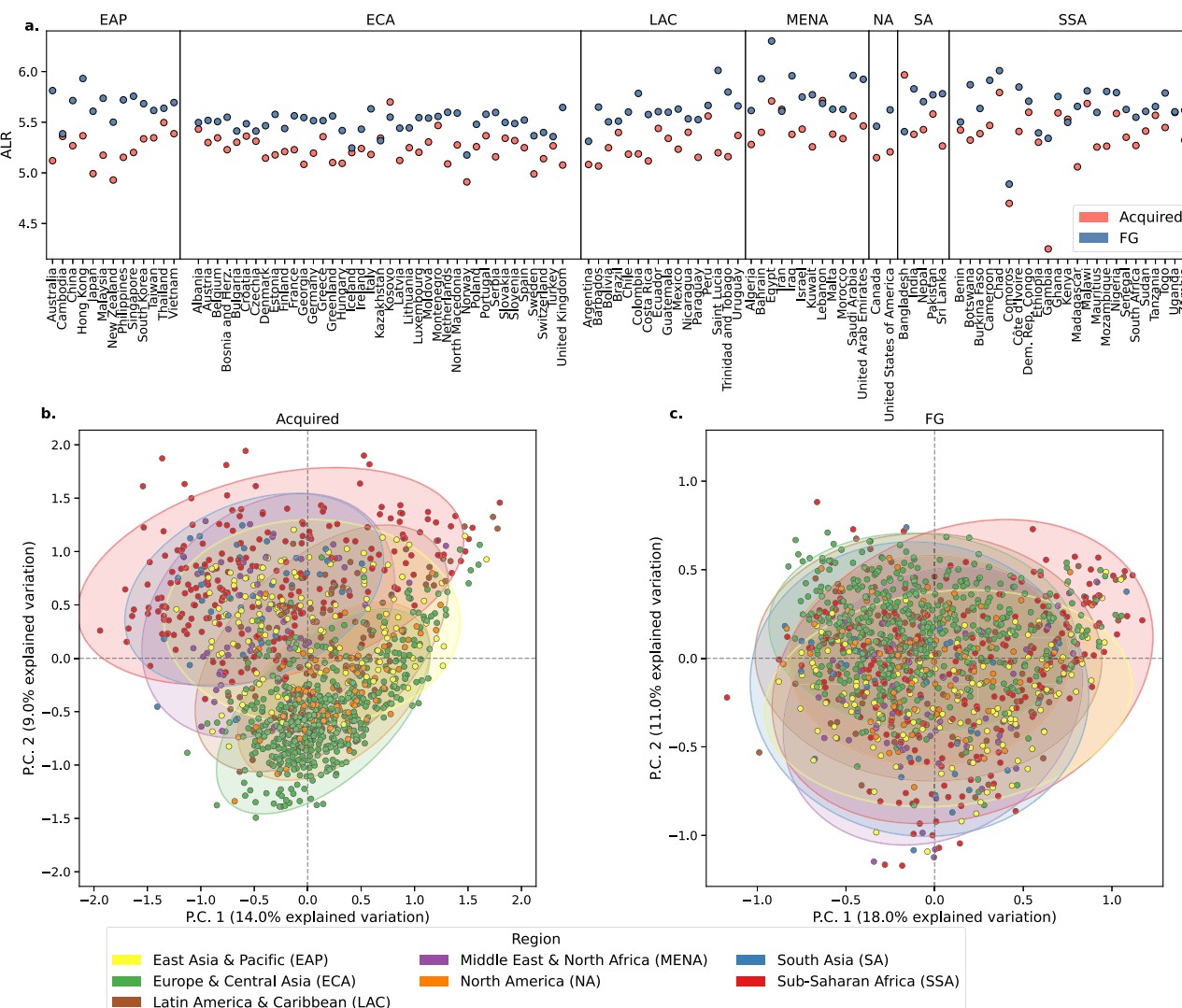

**Fig. 1 | Abundance and beta-diversity of ARGs across geographical regions.** **a** The country-wise ALR abundances of acquired ARGs and the FG ARGs. **b**, **c** PCA biplots of resistance genes (98% homology grouping), in which the PCA loadings were calculated from CLR values. Each marker represents a sewage sample and is colored by the world region for **b** acquired and **c** FG ARGs.

explained only 7.4% of the FG ARG resistome (permanova, p = 0.001, Fig. 1b). World regions could better explain the beta diversity of the combined FG collection than the two resources individually, 7.2% and 6.8% for ResFinderFG and Daruka, respectively (permanova, p = 0.001, Supplementary Fig. 7). Procrustes analyses indicated that the acquired resistome was slightly more closely related to the bacteriome (0.88, p = 0.001) than the FG resistome (0.69, p = 0.001).

Analyzing the resistomes at drug class levels showed that the abundances of the acquired ARGs were more consistent with world regions than for the FG ARGs, although there was a lot of unexplained variation (permanova, p = 0.001, Supplementary Table 1). An exception was glycopeptide resistance, where regions explained just 2.6% of the variance for acquired ARGs but 7.8% for the FG resistome (permanova, p = 0.001). Regions only explained a minor part of the beta-diversity of fluoroquinolone and polymyxin resistances for either ARG collections (Table 1, Supplementary Data 6).

### Urban sewage total resistome and ARG variants are subject to distance-decay effects
Exploiting that our sampling sites were taken at distances ranging from less than 1 km to >15,000 km, we wanted to determine the link between sample-wise distance and metagenomic similarity using linear distance-decay (DD) models (Fig. 2, Supplementary Table 2). We stratified sampling pairs based on whether they included different countries or even different regions. We observed significant distance-decay effects between the pairwise sample distance and resistome and bacteriome similarities, and that the rate of DD varied over local (within country), regional, and global scales (between regions).

However, contrary to our original hypothesis that the FG ARGs would have more dispersal limitations than the acquired ARGs, the DD models suggested that there is a significant difference in the DD slopes for acquired ARGs once outside a country (slope$_{within region}$ = −0.044, Supplementary Table 3). The DD slopes for the FG ARGs did not show a significant difference at the three spatial scales (Supplementary Table 3).

There was no detected effect of physical distance on taxonomic compositions at national or regional scales, but at the interregional scale, we observed a slight increase in similarity (reverse decay; slope=0.0015; P = 0.0). Mantel tests showed that the FG ARGs were slightly more correlated to bacterial dissimilarity (ρ = 0.76; P = 0.001) than the acquired ARGs (ρ = 0.7; P = 0.001)

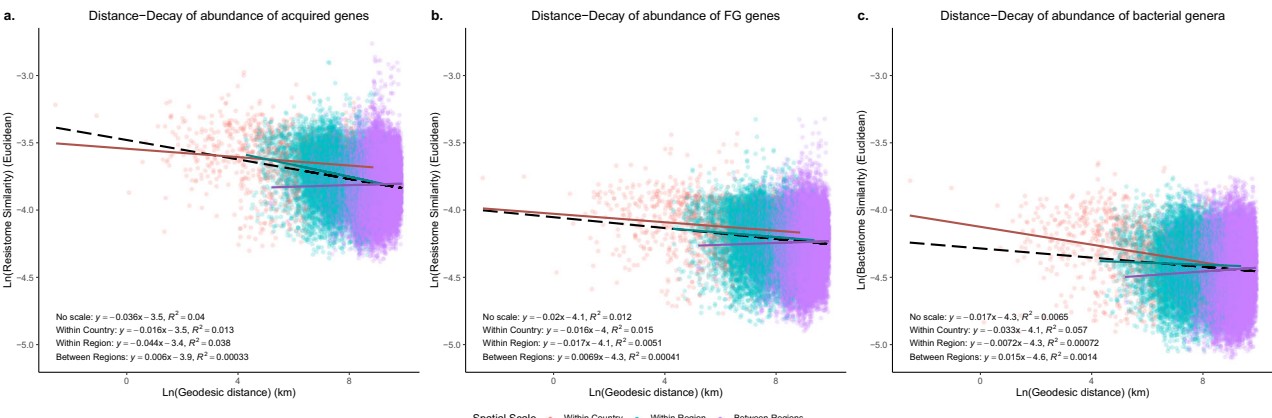

**Fig. 2 | Distance-Decay analyses of read abundances for city resistomes and bacteriomes.** Distance-Decay relationships for the bacteriome and resistome city communities ($n = 60{,}030$ pairwise comparisons) for the abundance of **a** Acquired ARGs, **b** FG ARGs, and **c** bacterial genera. The x-axis shows the pairwise city distances in kilometres (km), and the y-axis shows similarities. The dashed line represents the fit across all spatial scales and individual fits in solid: cities within the same country, cities within the same region, and cities that are in different regions. Model parameters and adjusted $R^2$ values are listed for each model in its corresponding plot.

We used metaSPAdes to recover contigs from the urban sewage samples to investigate the distance-decay patterns at the gene-variant level. We identified 1633 different ARGs in 83,869 contigs with the Flankophile pipeline. Among these, 28,482 contigs contained 452 different acquired ARGs, and 56,091 contigs contained 1181 different FG ARGs.

We found 27,217 variants of these known reference sequences, including 6098 variants of acquired genes (median: 3 variants/ARG, range: 1–269) and 21,119 FG ARGs (median: 4 variants/ARG, range: 1–325). Only 1060 of the ResFinder ARG variants and 3283 of the functionally identified ARG variants were observed more than once. Most of these were only found in a single or a few cities and countries, despite observing some variants in more than 300 cities (Supplementary Fig. 8).

For presence within and between regions, we observed an initial drop in prevalence in the countries, followed by an uptick in regional prevalence of different ARG variants (Supplementary Fig. S8a, b). Clustering cities based on acquired ARG variants resulted in strong regional grouping (Fig. 3a). However, clustering on FG variants did not (Fig. 3b). For example, cities in SSA and SA were more separated in their acquired variants from the rest of the world, except Pretoria, which clustered more with European and Central Asian cities.

Geographic distances between cities were strongly negatively correlated with ARG variant sharing (Fig. 3c, d; Supplementary Table 2). However, this effect completely disappeared at the inter-regional scale for acquired ARGs (slope = −0.01; $P = 0.11$). The FG genes behaved differently with distance, continuing to decay gene variant similarity sharing across regions (slope = −0.084; $P = 1.7 \times 10^{-102}$). Mantel tests over all the spatial scales suggested that physical distance was more correlated to acquired variant sharing ($\rho = 0.6$; $P = 0.001$) than to FG variant sharing ($\rho = 0.32$; $P = 0.001$, Supplementary Table 2).

The individual distance-decay relationships for each world region revealed that these effects mirror the global trends (Supplementary Fig. 9) but with greater variation (Supplementary Table 4), likely due to fewer data points available for specific regions (Supplementary Fig. 1). Interestingly, the acquired resistomes for East Asia & Pacific and SSA had some of the steepest declines in similarities between countries with slopes of −0.1 ($P < 0.001$, Supplementary Table 4), suggesting less connectivity between cities in those countries. Oppositely, there were only weak slopes for North American resistomes, indicating that despite large distances, the similarity of cities in North America does not decrease significantly.

## Network community detection reveals clustering of fecal bacteria and community clustering of acquired and FG resistomes

To elucidate the global dynamics of the sewage resistome, we constructed a correlation network integrating abundance data on bacterial species and ARGs (Supplementary Data Files 3–5). The network was reconstructed on a subset of elements consistently detected worldwide to effectively manage the typical sparsity of metagenomic data (details in Methods). This subset included 1865 mOTUs, 252 FG ARGs, and 71 acquired ARGs, with the latter tending to show a less uniform global spread.

The resulting network exhibited a sparse correlation structure with an edge density of ~1.1%, with ~25% of the edges connecting bacterial species and ARGs, and 857 (~40%) of the vertices were isolated, showing no relevant relationships. Furthermore, the network demonstrated a pronounced community structure with a modularity index of 0.7, indicating tightly interconnected distinct clusters of bacteria and ARGs likely originating from diverse ecological, environmental, and human-associated sources (Fig. 4). We observed that the six main communities detected in the network (Fig. 4b) contained 226 FG and 55 acquired ARGs and varied greatly in their number of elements, composition, and connectivity. Detailed plots of each community are in Supplementary Fig. 10.

The prevalence of human gut-associated species was noticeably high in communities 1 and 6, with 70% (115 species) in community 1 and 79% (49 species) in community 6. There were other notable differences in the composition of these two communities. In community 1, 78% of the human-associated microbiome species belonged to the phylum *Bacillota* (formerly known as *Firmicutes*), and other phyla included *Actinomycetota* (11%), *Pseudomonadota* (6%), *Euryarchaeota* (1%), and *Mycoplasmadota* (1%) (Supplementary Fig. 10a, Supplementary Fig. 12). In contrast, in community 6, 67% of the human-associated bacteria were members of the phylum *Bacteroidota* (formerly known as *Bacteroidetes*), with other phyla being *Pseudomonadota* (18%), *Bacillota* (8%), *Fusobacteria* (2%), and *Spirochaetota* (2%) (Supplementary Fig. 10f, Supplementary Fig. 12). The number of ARGs present in the two communities also differed greatly, as two acquired and 46 FG ARGs were part of community 1, and only two acquired and eight FG ARGs in community 6. Interestingly, 16 of the 46 FG genes in community 1 conferred resistance to glycopeptides (33%).

Community 2 showed low connectivity (around 2% internal edge density), especially compared to other communities (Supplementary Fig. 10b). It comprises 278 mOTUs, including 26 functional and 14 acquired ARGs. About 65% of its members are unassigned bacterial

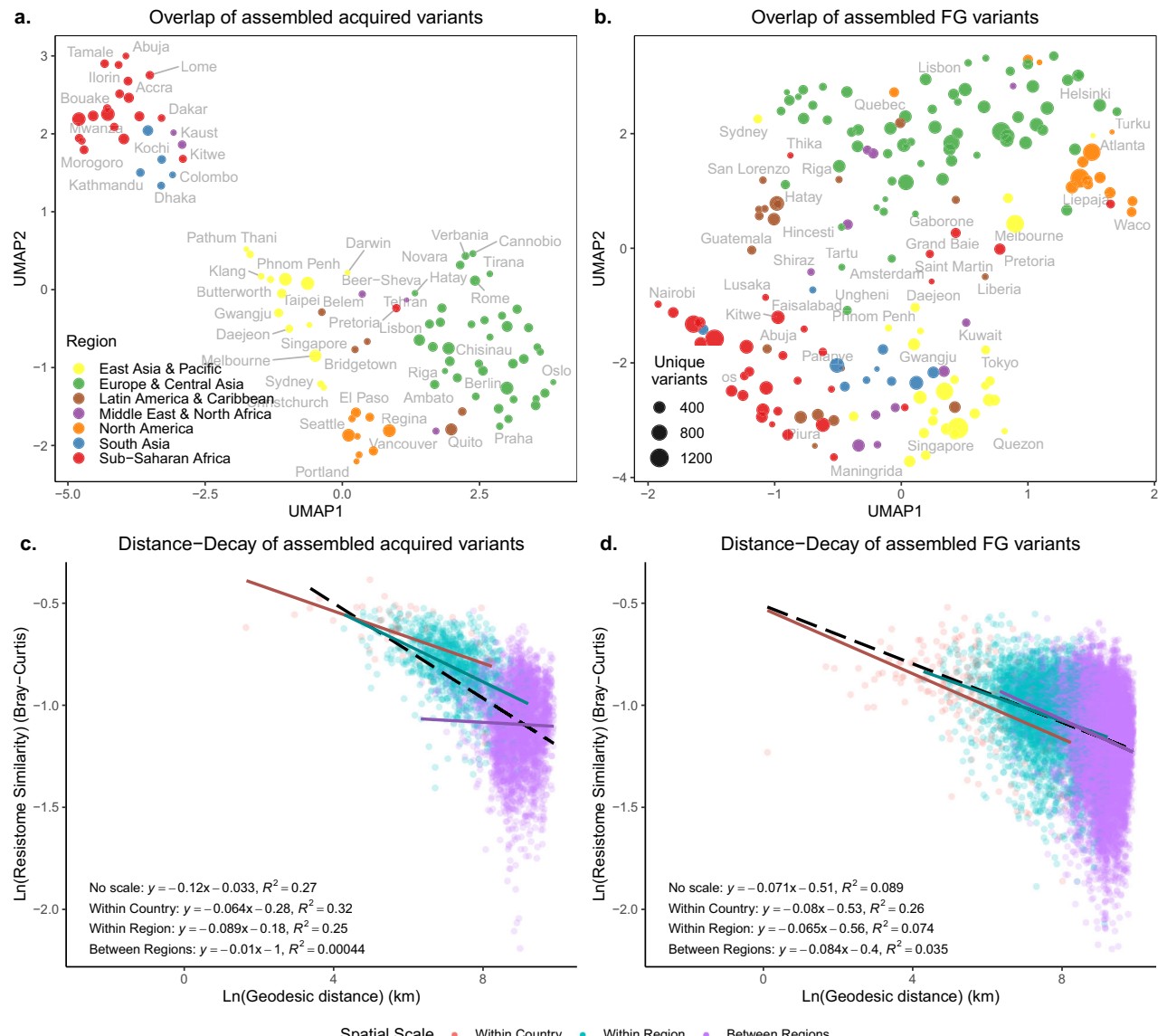

**Fig. 3 | UMAP and distance-decay analysis of resistomes.** UMAP clustering of shared variants among the cities for **a** acquired ARGs and **b** FG ARGs. Only cities with more than 100 non-singleton alleles were retained, and Hellinger transformed and clustered with the UMAP algorithm. Each marker represents a city, colored by region and sized by the number of unique variants in that city. City labels were optimized using the ggrepel package to minimize overlap. Distance-decay relationships for assembled city resistomes across different spatial scales for **c** acquired

($n = 4656$ pairwise comparisons) and **d** the FG variants ($n = 16,471$ pairwise comparisons). The x-axis shows the pairwise city distances in kilometers (km), and the y-axis shows resistome similarities. The dashed line represents the fit across all spatial scales and individual fits in solid: cities within the same country, cities within the same region, and cities that are in different regions (between regions). Model parameters and adjusted $R^2$ values are listed for each model in its corresponding plot.

species that serve as both abundant and central nodes. Among these, Bacteria *ext_mOTU_v3_18373* and Bacteria *ext_mOTU_v3_18384* exhibit particularly high connectivity and abundance, though their phyla remain unclassified. Beyond these two species, the other central taxa are also unidentified but span various families, including *Acidaminococcaceae, Atopobiaceae, Comamonadaceae, Eubacteriaceae, Moraxellaceae, Selenomonadaceae, Sporomusaceae,* and *Synergistaceae.* The only central, fully classified species are *Brachymonas denitrificans* and *Acinetobacter towneri.*

Community 3 contained the largest number of different acquired ARGs including macrolide resistance genes (*mef(c), mef(b), mph(e), mph(g),* and *msr(d)*); lincosamide resistance genes (*lnu(b)* and *lnu(d)*); and aminoglycoside resistance genes (*aadA* variants and *ant(6)-Ia*). This community also encompassed a diverse range of bacterial taxa, including *Acinetobacter, Chryseobacterium,* and *Flavobacterium,* as

well as some human-associated species such as *Acinetobacter johnsonii, Akkermansia muciniphila, Dialister invisus,* and *Lactococcus lactis.* Notably, it contained a large proportion of unassigned taxa (Supplementary Fig. 10c).

Community 4 was distinguished by its uniform composition and a very high prevalence of FG ARGs (Supplementary Fig. 10d). This community exclusively consisted of bacteria from the *Enterobacteriaceae* family (19 species) and comprised 126 nodes, ~85% of which were ARGs. Central to this community were *Escherichia coli* and *Klebsiella pneumoniae,* which, along with 17 other closely related species, formed a tightly knit group. These bacteria were associated with a diverse array of ARGs, specifically 104 FG ARGs.

Community 5 exhibited a coherent structure consisting of 113 elements, where 90 were bacteria and 23 ARGs (Supplementary Fig. 10e). This community predominantly included 65 bacteria from

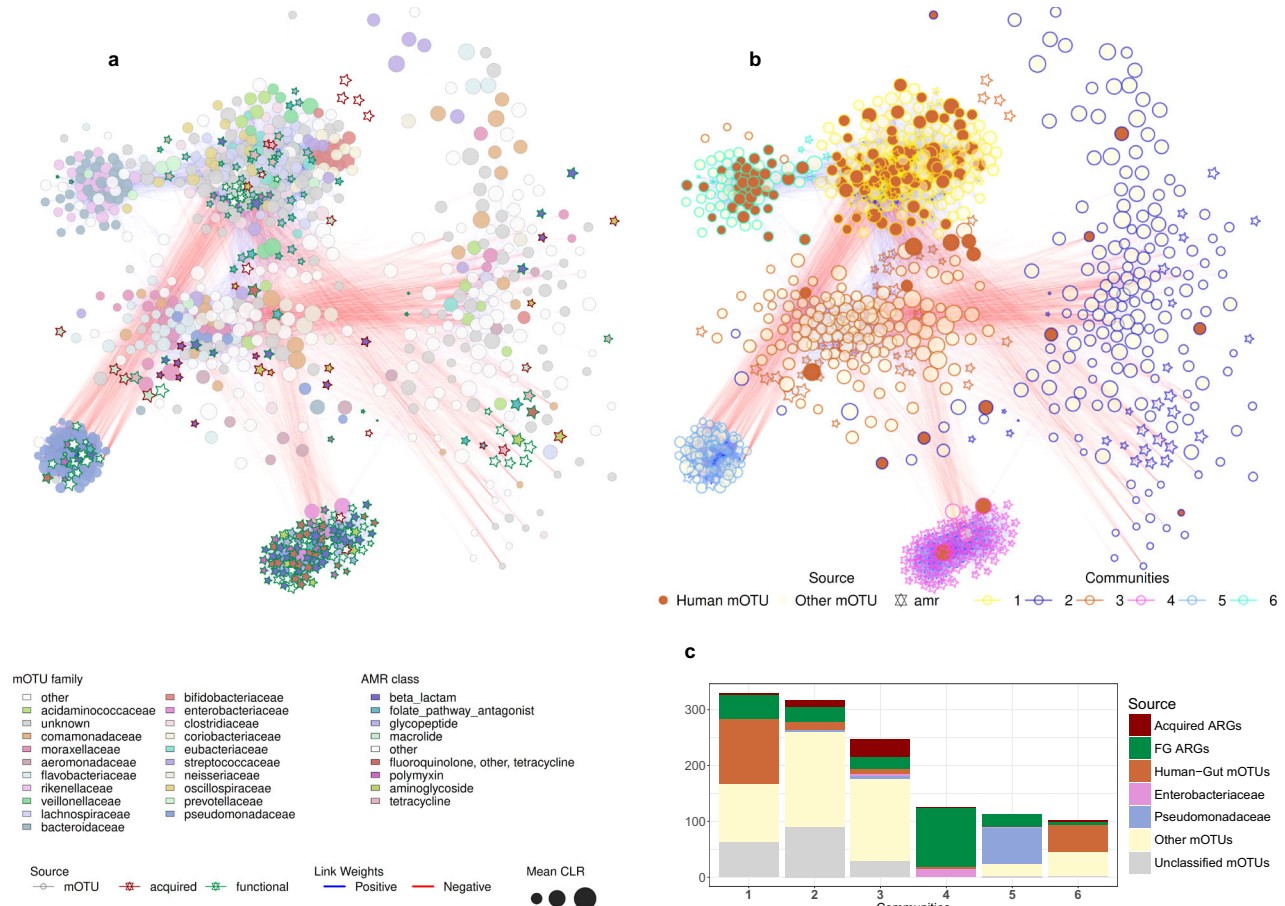

**Fig. 4 | The core of the correlation network of the abundances of bacteria and ARGs in the wastewater samples.** Each node represents either a species or an ARG, where an edge is the positive (blue) or negative (red) correlation between two nodes. The nodes are shaped by the database source (mOTU, acquired, or FG) and sized by the mean CLR value across samples. **a** Network graph showing the distribution of species and ARG, where a node is colored by either mOTU family or antimicrobial resistance class. **b** Network graph showing the communities detected. Nodes are here colored by whether they are an ARG or if the bacterial species is related to the human gut. The edge colors of the nodes highlight which community they were part of. **c** The count of the type of nodes in each community.

the *Pseudomonadaceae* family, seven from the *Moraxellaceae* family, and five from the *Yersiniaceae family*. The extraordinary internal connectivity of *Pseudomonadaceae*, with an edge density reaching a surprising 85%, defines the robust structure of this community. Additionally, the ARGs in this community were especially well-connected to the *Pseudomonadaceae* core, far more than to other families. These mainly functional ARGs conferred resistance to beta-lactams, fluoroquinolones, polymyxins, and tetracyclines.

## Discussion

Continuous monitoring of the distribution and dissemination of AMR across different environments remains a critical task[1,27], and we have previously shown how sewage sampling can effectively support such surveillance activities[15,16,28]. While multiple studies have shown how populations of bacteria in various environments can be carriers of ARGs[18,27,29], existing reference gene databases primarily contain ARGs encountered in clinical settings[11,14].

In this study, we sought to investigate the differences in diversity and abundance of the acquired ARGs from the ResFinder[12] database with two collections of ARGs identified with FG[13,14]. The culture-independent approach of FG offers an opportunity to identify ARGs in both environmental and human-associated microbial populations that have not been characterized before[30,31]. By analyzing our global collection of 1240 untreated wastewater samples from 111 countries

sampled between 2016 and 2021, we found that functionally identified ARGs (FG ARGs) were more globally dispersed in the sewage resistome than acquired ARGs. This suggests that novel resistance mechanisms might emerge in any geographical location and that possible global hotspots are not dependent on the genomic availability but rather on anthropogenic factors such as selection and transmission.

We confirmed our earlier wastewater studies suggested a regional separation of the acquired ARG abundances[15,16], but we also found that this regional signal appeared weaker in this study. Regions could only explain 12% of the beta diversity for the acquired ARGs and even less, at 7.4% for the FG ARGs. We decided to investigate the distance-decay (DD) patterns to elucidate whether the dissemination of ARGs faces significant barriers. Numerous studies have shown that the decay of similarity with geographical distances is a universal pattern observed across all domains of life[32–37]. Our models, based on both city-wise read abundances and the proportion of shared ARG variants, showed significant DD effects for both the acquired and FG ARGs. However, the slopes of these relationships varied across spatial scales. The acquired ARGs showed a strong negative correlation within a region, but the effect disappeared at inter-regional scales. This pattern could perhaps be due to regional-specific selective pressures related to population differences or competitive exclusion; if an ARG encoding a similar phenotype has already established itself at a specific site, another variant will be excluded. In contrast, the FG ARGs exhibited a more

gradual and continuous global decay, suggesting that while they are more globally dispersed, their dissemination might be shaped by other processes than physical distances, such as environmental conditions or ecological differences[32,33,38–40]. It should be noted that the observed DD effects might be influenced by the unequal representation of samples from the different regions, with most samples originating from Europe. The spatial patterns of ARGs may also reflect the underlying microbial community structure in the sewage systems, as we observed that FG ARGs were more linked to the bacteriome than the acquired ARGs. This suggests that the majority of FG ARGs were likely still embedded in their original genomic contexts and have not yet been mobilized, representing a latent reservoir of resistance.

To further investigate the bacteriome-resistome relationship, we used network analyses based on read abundances to identify six main communities of bacterial taxa correlated with ARGs. Two of these communities contained primarily taxa associated with human gut samples, confirming our recent study that it might be feasible to source attribute taxa and ARGs[17]. Interestingly, one was dominated by *Bacillota* (community 1) and the other one by *Bacterioidota* (community 6), two phyla that together account for over 90% of the abundance of the detected species in human gut microbiota[41,42], and their ratio can be a potential indicator of various physiological and pathological states[43–45]. Community 4 had the largest amount of FG ARGs, and central to this community were *Enterobacteriaceae* nodes, specifically *E. coli* and *K. pneumoniae*. These two in community 4 may partly reflect methodological biases, as both *E. coli* and *K. pneumoniae* have been used as bacterial hosts in functional metagenomic studies[13,46]. Community 3 had the highest count of acquired ARGs and contained both environmental and human-associated taxa, which could indicate multiple context locations. The presence of ARGs and human-gut-associated taxa in the detected network communities could indicate a potential for faster dissemination and align with the acquired DD slopes, as it has recently been shown that human-associated bacteria displayed faster dispersal rates[35].

Whether including FG ARGs in resistome studies and reference databases is clinically relevant remains an open question. On the one hand, FG offers a powerful way to explore the distribution of ARGs in non-pathogenic or non-culturable bacteria that have yet to be mobilized or selected for in clinical settings[14,47]. On the other hand, their inclusion may introduce too much noise into resistome analyses, as the FG ARGs could be part of the intrinsic resistome and just be a marker for the bacteria they belong to, thus posing low or no threat of becoming an issue to global health[48]. Careful interpretation is therefore required to avoid overestimating the risk associated with FG ARGs, particularly since some of the genes may only be functionally active under experimental settings and may serve other roles or remain inactive in natural microbial communities. While sewage sampling offers a snapshot of the resistome of a human population, it is worth noting that sewage also contains environmental and industrial inputs and is affected by the local climate and seasonality[15,17,47]. These non-human contributions to sewage, together with unequal sampling coverage of world regions, have likely introduced noise into the results. This noise is particularly relevant to keep in mind for our study of the FG ARGs, as we have seen evidence that some of the FG ARGs appear to be in environmental taxa. Further validation of the observed resistome patterns is necessary to elucidate the ecological and evolutionary contexts of ARG dissemination and their clinical relevance across environments beyond sewage.

Our findings demonstrate that the FG ARGs represent a latent reservoir of resistance located globally. The functional resistome was more associated with the bacteriome, suggesting a strong evolutionary barrier to their mobilization. Our identified distance-decay and network communities suggested that differential selection and niche competition, rather than dispersal, shape the global sewage resistomes. The acquired ARGs likely reflect most of the current burden of resistance. In contrast, the FG ARGs may become a future problem; thus, including ARGs identified with FG in routine surveillance programs could serve as an early warning system for their mobilization.

## Methods

### Sampling and sequencing

We started by retrieving the collection of metagenomic reads and assemblies in our previous study[15]. We repeated our global calls, recruited partners, and received additional untreated sewage samples from 2018 to 2021. The sampling, DNA extraction, library preparation, and sequencing methods were identical to those used in our previous work[15,16].

In brief, bottles of untreated sewage were frozen by the partners and shipped to Denmark. The untreated sewage was thawed, and 250 mL of each sample was centrifuged at $10,000 \times g$ for 10 min to retain the pelleted bacteria. DNA was extracted using our previously published modified DNA extraction protocol[49], and low-concentration samples were vacuum-concentrated. All samples were shipped on dry ice for sequencing at Admera Health (New Jersey, USA). Here, KAPA Hyper PCR-free library preparation was used, and sequencing was carried out on the Illumina NovaSeq6000, targeting >35 M clusters, with paired-end (2) sequencing and 150 cycles per end. We thus aimed for at least 35 M * 2 ends * 150 bp = 10.5 Gbp of sequence per sample for comparability with previous studies. The cities from which sewage samples were retrieved are marked in Supplementary Fig. 1, with the number of samples from each country and sampling year. A combined list of samples, metadata, sequence data accessions, and other relevant information can be found in Supplementary Data 1.

### Bioinformatic processing of reads with ARGprofiler

We processed the raw FASTQ sewage metagenomes using our recently published pipeline, ARGprofiler[26] v1.0.0, with default settings. This pipeline was designed with snakemake 7.30.1[50,51] to profile ARGs both quantitatively and qualitatively in metagenomic datasets. In brief, ARGprofiler takes the raw FASTQ reads and begins by quality checking and trimming them with fastp[52] 0.23.2. Settings for fastp were set as follows: overlap_diff_limit = 1, average_qual = 20, length_required = 50, --cut_tail. After trimming, ARGprofiler uses KMA[53] 1.4.12a to do global alignment of the trimmed reads against the two reference databases, mOTUs[54] (v3.0.3) and PanRes[26] v1.0.1. The following alignment parameters were used: 1, −2, −3, −1 for a match, mismatch, gap opening, and gap extension. For pairing of reads, we used a value of 7 and a minimum relative alignment score of 0.75.

Additionally, all the new sequence runs were subjected to metagenomic assembly, following the same approach as in the previous study[15,16]. Metagenomic assembly was done using metaSPAdes[55] (SPAdes v. 3.13.0) with k-mer sizes: 27, 47, 67, 87, 107, 127 and the pre-correction flag set. We filtered away contigs shorter than 1 Kbp due to their large numbers and lack of synteny information, which is important for epidemiological inference[15].

**Reference sequence databases.** We used the PanRes database as the reference collection for ARGs, as it incorporates multiple existing databases of ARGs into a single non-redundant collection, totaling 14,078 unique ARGs[26]. This study focused on three of the included source databases: ResFinder[12], ResFinderFG 2.0[14], and those from Daruka et al.[8] (tagged as csabapal in PanRes). The latter two databases are products of FG. Throughout this study, we group the references based on their presence in ResFinder or the functional collections and refer to them as acquired or FG respectively. The few ARGs in both groups were uniquely counted as acquired for the purpose of this study (Supplementary Fig. 1).

We homology reduced all PanRes reference sequences using USEARCH[56] (v.11.0.66) to 98% nucleotide identity, with query and target coverage thresholds of 98%. To maintain more alleles in the

acquired and FG groups, we assigned cluster representatives as acquired if they had at least one cluster member tagged as acquired and cluster representatives as FG if they had an FG member and no acquired cluster members (Supplementary Data 2).

ARGprofiler uses the mOTUs database[54,57] (v3.0.3) with the KMA aligner[53] (v. 14.12.a) to profile the metagenomes. To deal with many mOTUs having a placeholder name (orphans), we padded the missing names with the highest known taxonomic ranks. All resulting KMA mapstat files with mOTUs sequences were filtered to retain only alignments to bacteria. For analyses of beta-diversities, we filtered away taxonomies that made up less than 0.0001% of the reads assigned to mOTUs across all metagenomes.

## Quantifying ARGs using compositional methods

We reported the total number of read fragments aligned to the three reference sequence sets (mOTU, Acquired, and FG), as well as the number of unique reference hits and Shannon[58] diversity for each sewage sample.

Estimation of the regional pan- and core-resistomes was done as follows: the number of unique ARG references hit in samples from a region was calculated as the size of the regional pan-resistome. The core-resistome was then estimated as the number of unique ARGs that were observed in at least 50% of the regional samples. The pan- and core-resistomes were then stratified by the AMR class for each collection (acquired and FG), and the subset of those classes in both collections was plotted.

KMA reports the count of read fragments aligned to reference sequences, which we treat as parts of a composition[59]. We applied the additive log-ratio transformation (ALR) to length-adjusted ARG read fragment counts with the sum of bacterial mOTUs reads as the reference component.

To perform statistical analysis on ARG abundances, we imputed zeroes following procedures highlighted in previous work[59]. The length-adjusted and zero-replaced abundance counts were then transformed with the centered log-ratio (CLR) transformation, which uses the geometric mean as the reference component[60–62].

We compared sample beta diversities using Principal Component Analysis (PCA), following the methodology outlined in previous work[60–62]. The zero-imputed and length-adjusted counts were centered by the geometric mean, scaled by the total log-ratio variance, transformed into CLR values, and eigen-decomposed to obtain eigenvectors and eigenvalues. These were then used to calculate the principal components and visualized in biplots. In Python 3.12[63], the pycodamath[64] package was used for the compositional analyses, and visualization of abundances and diversities was created using matplotlib 3.8.2[65], seaborn 0.13.2[66], and geopandas 0.14.3[67].

In R 4.3.2[68], the Procrustes function from the vegan 2.6[69] was used to calculate the degree to which resistome and bacteriome PCA scores correlated, and the adonis2 function was used to determine the proportion of beta-diversity variance attributable to World Bank Regions.

## Metagenomic ARG variants

Using the metagenomic assembled scaffolds produced by metaSPAdes (SPAdes v. 3.13.0), we searched for ARG variants from ResFinder and the functional ARG collections with the Flankophile tool[68] (v. 0.2.8). Briefly, the pipeline searches assemblies for ARGs, extracts and clusters them, and calls new variants.

Using the annotations of which ARGs and identified variants were found across the assemblies, we created contingency tables summarizing observed occurrences of the ARG sequences (closest known reference and specific variant) across geographical groupings (region, country, and city). ARG sequences that were only observed once and geographical areas with less than 100 ARG copies observed were discarded. Each sample in the contingency tables was normalized using

the Hellinger transformation[70], as implemented in the package vegan 2.6[69] for R 4.3.2[68].

Unimap Manifold Approximation and Projection (UMAP)[71], as implemented in the R package umap 0.2.10.0[72], was applied to the normalized contingency matrices to reduce their dimensionality, allowing a projection of the number of closest references and called variants shared between the different cities.

## Estimating distance-decay relationships

The distance-decay effects were investigated by comparing physical distances with resistome similarities. We calculated the rate of distance-decay as the slope of linear regression on the relationship of ln-transformed similarities with ln-transformed distances, as outlined in Martiny et al.[32] and Nekola and White[33]. The physical distance between cities was obtained by calculating geodesic distances based on latitudes and longitudes with the R package sf 1.0.15[73]. Abundance similarities were calculated as Aitchison distances (Euclidean distances on CLR values) with the simil function from the proxyC 0.4.1 package[74]. Similarities of the assembled variant resistomes were calculated using Bray-Curtis dissimilarities[75] on the contingency tables as implemented in the vegdist function in vegan 2.6[69].

Distances and dissimilarities were log-transformed, and linear models were fitted both for comparing all samples and within stratified groups of comparisons: those with cities within the same country, cities within the same region, and cities from different world regions. The slopes of the lines for the three different spatial groupings were compared with the emtrends function from emmeans 1.10.5[76]. Mantel tests[77] were applied to measure the correlation of the physical and resistome distance matrices (not ln-transformed) with 999 permutations, as implemented in the mantel function in vegan 2.6[69].

## Network analyses

A shared correlation network was constructed from mOTU and ARG abundance data. To increase the signal-to-noise ratio and reduce sparseness, we imposed a 25% sample presence prevalence threshold for mOTUs and a median adjusted measure ≥1.

For ARGs, due to variable sample depths (as shown in Supplementary Fig. 11), we excluded samples with total abundances below the 10th percentile. Subsequently, we applied a 25% prevalence threshold and a median of non-zero abundances ≥0.01.

After these filtering steps, we preserved only samples that retained <90% of the abundance for mOTUs and ARGs. This led to 912 samples organized into two matrices: 1865 mOTUs and 323 AMR genes (252 FG, 71 Acquired).

We applied CLR transformations, replacing zeros with a pseudo-count (65% of the detection limit)[78], merged the normalized data, calculated Spearman correlations, and derived an adjacency matrix using an absolute threshold of ≥0.5. The resulting network is defined as undirected, with weighted and signed links where each node represents an mOTU or an ARG, and the links represent the preserved correlations between them. Finally, we applied a signed version of the modularity algorithm[79], capable of handling negative link weights, to detect communities within the network.

To identify the human microbiota, we downloaded the Unified Human Gastrointestinal Genome collection (UHGG) (v2.0.2)[80] and matched the species from this database to those in our classified communities by their clustered names to determine the number of human gut-associated mOTUs. The entire analysis was conducted in R using the mgnet package v0.3-beta (https://github.com/Fuschi/mgnet).

## Reporting summary

Further information on research design is available in the Nature Portfolio Reporting Summary linked to this article.

## Data availability

The raw sequencing reads and metagenomic assemblies have been deposited at the European Nucleotide Archive for the different rounds of sampling, which are available under project accession numbers: PRJEB40798, PRJEB40816, PRJEB40815, PRJEB27621, and PRJEB84064. The processed data such as KMA mapstat files, count matrices, and flankophile output, have been deposited on Zenodo at https://doi.org/10.5281/zenodo.14652832. The metadata associated with the samples and reference genes are in the Supplementary Data Files 1, 2 and network data in Supplementary Data Files 3, 5. Source data are provided with this paper. This study also utilized the Natural Earth dataset to plot geographical shapes and the publicly available databases PanRes, mOTUs and the UHGG catalog. Source data are provided with this paper.

## Code availability

All code used for analysis and data visualizations has been deposited on GitHub at https://github.com/genomicepidemiology/gs3_acquired_vs_FG[81].

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

## Acknowledgements

We want to extend our gratitude to everyone who has helped with sampling, shipping, and the logistics of transporting sewage to Denmark. A special thanks to the laboratory technicians at DTU that has organized the sample collection, material transfer, sequencing and logistics. Lastly, we would like to thank the Novo Nordisk Foundation (Grant: NNF16OC0021856: Global Surveillance of Antimicrobial Resistance) and the European Union's Horizon H2020 research and innovation programme (Grant: No. 874735) for funding the work.

## Author contributions

F.M.A. and P.M. conceived the study, and F.M.A. and M.K. secured the funding. H.M.M., P.M., and N.P. did quality control and sample selection. Trimming, mapping and assembly were performed by N.P., H.M.M., P.M., A.F., and A.B. carried out data and statistical analyses and visualization. H.M.M. and P.M. drafted the initial manuscript with input from A.F., A.B., C.B., D.G.J.L., M.K., D.R., I.C., and F.M.A. Members of the global sewage consortium carried out sewage sampling, filled in metadata and shipped the samples to DTU. All authors helped to review and improve the manuscript.

## Competing interests

The authors declare no competing interests.

## Additional information

Hannah-Marie Martiny or Patrick Munk.

## Global Sewage Consortium

Hannah-Marie Martiny [1,123] ✉, Patrick Munk [1,123] ✉, Nikiforos Pyrounakis[1], Christian Brinch [1], Frank M. Aarestrup [1], Alessandro Fuschi [2], Daniel Remondini [2], Ágnes Becsei[3], István Csabai[3], D. G. Joakim Larsson[4], Marion Koopmans[5], Nawel Zaatout[6], Catherine Rees[7], Guangming Jiang[8], Jiahua Shi[9], Bernhard Benka[10], Franz Allerberger[10], Sandra Koeberl-Jelovcan[10], Khalil Hasan Abdulla[11], Ali Bin Thani[12], Anowara Begum[13], Carlon Worrell[14], Tamegnon Victorien Dougnon[15], Freddy Soria[16], Natasa Mazalica[17], Teddie Rahube[18], Andreza Francisco Martins[19], Carlos Alberto Tagliati[20], Larissa Camila Ribeiro de Souza[20], Ariane Nzouankeu[21], Muhammad Attiq Rehman[22], Jeff Gauthier[23], Roger C. Levesque[23], Sean D. Workman[24], Christopher Yost[24], Aiko Adell Nakashima[25], Andres Opazo[26], Gerardo Gonzales[26], Yongjie Yu[27], Pilar Donado-Godoy[28], Tadjidine Youssouf[29], Pablo-César Rivera-Navarro[30], Matijana Jergovic[31], Jasna Hrenovic[32], Renata Karpiskova[33], Julien Kalpy Coulibaly[34], William Calero-Cáceres[35], Mohamed Abouelnaga[36], Anna-Maria Hokajärvi[37], Annamari Heikinheimo[38], Soizick Le Guyader[39], Andreas Nitsche[40], Annika Brinkmann[40], Sara Schubert[41], Thomas Berendonk[41], Uli Klümper[41], Ernest Bonah[42], Solomon Asante Sefa[43], Andrew Camilleri[44], Courage Kosi Setsoafia Saba[44], Victoria Bernice Sedor[45], Kassiani Mellou[46], Theologia Sideroglou[46], Charalampos Kotzamanidis[47], Jens-Peter B. Henriksen[48], Mie Møller[49], Thorunn Rafnar Thorsteinsdottir[50], A. A. Mohamed Hatha[51], Sima Mohammad[52], Burhan Shamurad[53], Hiwa Ali Faraj[53], Dearbháile Morris[54], Louise O'Connor[54], Jacob Moran-Gilad[55], Roberta Orletti[56], Giuseppina La Rosa[57], Marcello Iaconelli[57], Antonio Battisti[58], Lucia Decastelli[59], Maira Napoleoni[60], Alessandra De Cesare[61], Gianluca Corno[62], Jelena Avsejenko[63], Ghassan M. Matar[64], Christian Penny[65], Luc Hervé Samison[66], Daniel L. Banda[67], Heera Rajandas[68], Sivachandran Parimannan[68], Malcolm Vella Haber[69], Sunita J. Santchurn[70], Julian Carrillo-Reyes[71], Julian Osvaldo Sanchez-Lara[71], René Arredondo-Hernández[71], Aleksandar Vujacic[72], Dijana Djurovic[72], Pushkar Pal[73], Carlos J. A. Campos[74], Isabelle Pattis[75], Stephen Chambers[76], Gert-Jan Jeunen[77], Charles Elikwu[78], Olayinka Osuolale[79], Mwapu Ndahi[80], Oluwadamilola Abiodun-Adewusi[81], Amen Ekhosuehi[82], Ayorinde Afolayan[83], Kayode Fashae[83], Mabel Kamweli Aworh[83], Akeem Olayiwola Ahmed[84], Ibrahim Adisa Raufu[84], Ismail Odetokun[84], Samaila Musa Chiroma[85], Claire Mitchell[86], Rune Holmstad[87], Natalie Weiler[88], Dariusz Wasyl[89], Magdalena Zając[89], Eugénia Cardoso[90], Célia Manaia[91], Maria Jorge Campos[92], Hor-Gil Hur[93], Min Joon Song[94], Sukhwan Yoon[94], Olga Burduniuc[95], Peiying Hong[96], Vladimir Radosavljevic[97], Angel Anika Cokro[98], Dagmar Gavacová[99], Marija Trkov[100], Rukayya Hussain Abubakar[101], Karen Keddy[102], Isabel Martínez-Alcalá[103], Marta Cerdà-Cuéllar[104], Ana Cañas[105], Fernando Gonzalez-Candelas[106], Yahia Ali Sabiel[107], Tanja van der Heijden[108], Yu-ping Hong[109], Julius John Medardus[110], Rajeev P. Nagassar[111], Cemíl Kürekci[112], Patrick McNamara[113], Lisa Wong[114], Erica Fuhrmeister[115], John Scott Meschke[116], Nicola Beck[116], Ayella Maile-Moskowitz[117], Jens Thomsen[118], Ouidiane Obaidi[119], Bruce Paterson[120], Anne Leonard[121], Lihong Zhang[121] & Kevin Chau[122]

[6]University of Batna 2, Batna, Algeria. [7]Melbourne Water Corporation, Melbourne, VIC, Australia. [8]School of Civil, Mining, Environmental, and Architecture Engineering, University of Wollongong, Wollongong, NSW, Australia. [9]University of Wollongong, Wollongong, NSW, Australia. [10]Austrian Agency for Health and Food Safety (AGES), Vienna, Austria. [11]The Supreme Council for Environment (SCE) in Bahrain, Manama, Bahrain. [12]University of Bahrain, Sakhir, Bahrain. [13]University of Dhaka, Dhaka, Bangladesh. [14]Environmental Protection Department, Bridgetown, St. Michael, Barbados. [15]Polytechnic School of Abomey-Calavi, Abomey-Calavi, Benin. [16]Universidad Católica Boliviana San Pablo, La Paz, Bolivia. [17]Public Health Institute of the Republic of Srpska, Banja Luka, Bosnia

and Herzegovina. [18]Botswana International University of Science and Technology, Palapye, Botswana. [19]Federal University of Rio Grande do Sul, Porto Alegre, Brazil. [20]Universidade Federal de Minas Gerais, Belo Horizonte, Brazil. [21]Centre Pasteur du Cameroun, Yaoundé, Cameroon. [22]Research and Productivity Council, Fredericton, NB, Canada. [23]Université Laval, Québec, QC, Canada. [24]University of Regina, Regina, SK, Canada. [25]Escuela de Medicina Veterinaria, Facultad de Ciencias de la Vida, Universidad Andrés Bello, Santiago, Chile. [26]Universidad de Concepción, Concepción, Chile. [27]Nanjing University of Information Science and Technology, Nanjing, China. [28]Global Health Research Unit on Genomic Surveillance of Antimicrobial Resistance (GHRU), CI Tibaitatá, Corporación Colombiana de Investigación Agropecuaria (AGROSAVIA), Funza, Colombia. [29]Laboratoire National de Référence Hospitalier, Moroni, Comoros. [30]University of Costa Rica, San José, Costa Rica. [31]Andrija Stampar Teaching Institute of Public Health, Zagreb, Croatia. [32]University of Zagreb, Zagreb, Croatia. [33]Veterinary Research Institute, Brno, Czechia. [34]Institut Pasteur de Côte d'Ivoire, Abidjan, Côte d'Ivoire. [35]UTA-RAM-One Health, Group for Universal Advance in bioScience, Department of Food and Biotechnology Science and Engineering, Universidad Técnica de Ambato, Ambato, Ecuador. [36]Suez Canal University, Ismailia, Egypt. [37]Tampere University, Tampere, Finland. [38]University of Helsinki, Helsinki, Finland. [39]French Institute Search Pour L'exploitation De LaMer (Ifremer), Nantes, France. [40]Robert Koch Institute, Berlin, Germany. [41]Institute of Hydrobiology, Technische Universität Dresden, Dresden, Germany. [42]Food and Drugs Authority, Accra, Ghana. [43]Public Health Laboratory, Ghana Health Service, Takoradi, Ghana. [44]University for Development Studies, Tamale, Ghana. [45]Veterinary Services Department, Ministry of Food and Agriculture, National Food Safety Laboratory, Accra, Ghana. [46]National Public Health Organisation (EODY), Athens, Greece. [47]Veterinary Research Institute of Thessaloniki, Hellenic Agricultural Organisation-DEMETER, Thermi, Greece. [48]Kommuneqarfik Sermersooq, Nuuk, Greenland. [49]Peqqik, Nuuk, Greenland. [50]University of Iceland, Reykjavik, Iceland. [51]Cochin University of Science and Technology, Cochin, India. [52]Shahid Beheshti University, Tehran, Iran. [53]The University of Sulaimani, Sulaymaniyah, Iraq. [54]National University of Ireland, Galway, Ireland. [55]BenGurion University of the Negev and Ministry of Health, Beer-Sheva, Israel. [56]Ambiente, Marche, Italy. [57]Istituto Superiore di Sanità, Rome, Italy. [58]Istituto Zooprofilattico Sperimentale del Lazio e della Toscana, Rome, Italy. [59]Istituto Zooprofilattico Sperimentale del Piemonte, Liguria e Valle d'Aosta, Italy. [60]Istituto Zooprofilattico Sperimentale dell'Umbria e delle Marche "Togo Rosati", Perugia, Italy. [61]University of Bologna, Bologna, Italy. [62]Water Research Institute, National Research Council (CNR), Verbania, Italy. [63]Institute of Food Safety, Riga, Latvia. [64]American University of Beirut, Beirut, Lebanon. [65]Luxembourg Institute of Science and Technology, Belvaux, Luxembourg. [66]University of Antananarivo, Centre d'Infectiologie Charles Mérieux, Antananarivo, Madagascar. [67]University of Malawi, Blantyre, Malawi. [68]AIMST University, COMBio, Kedah, Malaysia. [69]Environmental Health Directorate, St. Venera, Malta. [70]University of Mauritius, Reduit, Mauritius. [71]Universidad Nacional Autónoma de México, Mexico City, Mexico. [72]Institute for Public Health Montenegro, Podgorica, Montenegro. [73]Agriculture and Forestry University, Kathmandu, Nepal. [74]Cawthron Institute, Nelson, New Zealand. [75]Institute of Environmental Science and Research Limited (ESR), Christchurch, New Zealand. [76]University of Otago, Christchurch, New Zealand. [77]University of Otago, Dunedin, New Zealand. [78]Babcock University Teaching Hospital, Ilishan-Remo, Nigeria. [79]Elizade University, Ilara-Mokin, Nigeria. [80]Federal Ministry of Agriculture and Rural Development, Abuja, Nigeria. [81]Nigeria Field Epidemiology Training Program, Abuja, Nigeria. [82]University of Benin, Benin City, Nigeria. [83]University of Ibadan, Ibadan, Nigeria. [84]University of Ilorin, Ilorin, Nigeria. [85]University of Maiduguri, Maiduguri, Nigeria. [86]Department of Agriculture, Environment and Rural Affairs, Northern Ireland, UK. [87]VEAS, Slemmestad, Norway. [88]Departamento de Bacteriologia, Laboratorio Central de Salud Publico, Asunción, Paraguay. [89]National Veterinary Research Institute, Pulawy, Poland. [90]Águas de Portugal, SGPS, S.A, Lisboa, Portugal. [91]Centre for Biotechnology and Fine Chemistry (CBQF), Universidade Católica Portuguesa, Porto, Portugal. [92]Escola Superior de Turismo e Tecnologia do Mar (ESTM), Instituto Politécnico de Leiria (IPLeiria), Peniche, Portugal. [93]Gwangju Institute of Science and Technology, Gwangju, Republic of Korea. [94]Korea Advanced Institute of Science and Technology, Daejeon, Republic of Korea. [95]National Agency for Public Health, Chișinău, Republic of Moldova. [96]King Abdullah University of Science and Technology, Thuwal, Saudi Arabia. [97]Institute of Veterinary Medicine of Serbia, Belgrade, Serbia. [98]Nanyang Technological University, Singapore, Singapore. [99]Public Health Authority of the Slovak Republic, Bratislava, Slovakia. [100]National Laboratory of Health, Environment and Food, Ljubljana, Slovenia. [101]Faculty of Veterinary Sciences, University of Pretoria, South Africa. [102]Independent consultant, Johannesburg, South Africa. [103]Catholic University of Murcia, Murcia, Spain. [104]Centre de Recerca en Sanitat Animal (CReSA), IRTA-UAB, Universitat Autònoma de Barcelona, Bellaterra, Spain. [105]Instituto de Salud Carlos III, Madrid, Spain. [106]Universidad de Valencia, Valencia, Spain. [107]Sudan University of Science and Technology, Khartoum, Sudan. [108]Ara Region Bern AG, Herrenschwanden, Switzerland. [109]Centers for Disease Control, Taipei, Taiwan, ROC. [110]Sokoine University of Agriculture, Morogoro, Tanzania. [111]Eastern Regional Health Authority (ERHA), Sangre Grande, Trinidad and Tobago. [112]Mustafa Kemal Üniversitesi, Hatay, Turkey. [113]Marquette University, Milwaukee, WI, USA. [114]Massachusetts Water Resources Authority, Boston, MA, USA. [115]Tufts University, Medford, MA, USA. [116]University of Washington, Seattle, WA, USA. [117]Virginia Tech, Blacksburg, VA, USA. [118]Abu Dhabi Public Health Center, Abu Dhai, United Arab Emirates. [119]Dubai Municipality, WWTP Al Aweer, Dubai, United Arab Emirates. [120]Scottish Environment Protection Agency, Stirling, UK. [121]University of Exeter Medical School, Cornwall, UK. [122]University of Oxford, Oxford, UK.

