## [Transparent Peer review file · Nature Communications]

Geographics and bacterial networks differently shape the acquired and latent global sewage resistomes

Corresponding Author: Dr Hannah-Marie Martiny

Version 0:

Reviewer comments:

Reviewer #1

(Remarks to the Author)

Please see attached comments

(Remarks on code availability)

Reviewer #2

(Remarks to the Author)

This manuscript by Martiny, Munk, Fuschi, et al., as part of the Global Sewage Consortium, investigates the global distribution patterns of antimicrobial resistance genes (ARGs) in urban sewage, distinguishing between ARGs known to be mobilized (acquired ARGs) and non-mobilized genes identified via functional metagenomics (FG ARGs). The authors leverage an extensive dataset comprising metagenomic sequencing data from 1,240 samples collected from 351 cities across 111 countries between 2016 and 2022. The study employs robust bioinformatic pipelines, statistical modeling, and network analyses to elucidate the underlying ecological and geographical factors influencing ARG distribution.

The authors should be commended for undertaking this massive collaborative global effort, which has already provided key insights into the role of wastewater as a powerful tool for antimicrobial resistance (AMR) surveillance worldwide.

However, to this reviewer, the results in this manuscript represent a continuation of two previous publications from this same consortium (doi: 10.1038/s41467-019-08853-3 and 10.1038/s41467-022-34312-7) in describing the global distribution of AMR genes, albeit with an expanded sample set and geographical coverage. The novelty in this work primarily lies in the network analysis and the differentiation between acquired and functional metagenomic resistance genes. This analysis highlights distinct dissemination patterns between these two classes and demonstrates that the resistome in sewage is largely shaped by its microbial composition. While confirming that this pattern applies globally is interesting, previous studies have already shown similar microbiome-driven resistome structuring in human and wastewater samples (e.g., doi: 10.1038/nature17672).

Key results: The authors report significant differences between acquired and FG ARGs in their global distribution and abundance. The acquired resistome displayed strong geographical structuring, closely following regional distinctions, whereas the FG resistome showed more even global distribution. Additionally, clear distance-decay effects were demonstrated, with acquired ARGs primarily showing decay at national and regional scales, whereas FG ARGs exhibited global-scale dispersal limitations. Network analysis provided evidence that FG ARGs have stronger associations with bacterial taxa than acquired ARGs, suggesting different mechanisms of transmission and evolutionary pressures.

Validity: The data presented are technically sound, benefiting from a robust and well-documented methodology. Sequencing and analytical pipelines, such as ARGprofiler, metaSPAdes assemblies, and statistical analyses using compositional methods and network analyses, are appropriate and widely used in the field. Using compositional data analysis (CLR and ALR transformations) strengthens the validity of the findings. However, further validation of the functional ARG annotations,

possibly by additional experimental or comparative genomic methods, would be beneficial.

Significance: The study advances our understanding of ARG dissemination at a global scale, clearly differentiating between ARGs based on their mobilization histories. These insights are critical for informing global surveillance efforts and policies to mitigate antimicrobial resistance.

Data and methods: The extensive global dataset and robust bioinformatics pipeline enhance the quality and reliability of the findings. Supplementary materials adequately complement the main findings, but the supplementary data files could benefit from clearer metadata annotations or indexing to facilitate independent verification and reuse.

Clarity and context: Overall, the manuscript is clearly written and well-organized. The findings are appropriately contextualized within existing literature. However, the authors could improve clarity by summarizing key methodological steps at the beginning of the results section to guide readers through complex analyses. Additionally, certain figures (e.g., PCA plots and UMAP clustering) could be visually optimized for better readability, especially given the extensive geographical data.

Suggested improvements:

- The terms 'acquired' and 'functional metagenomic' for the two classes of genes are somewhat confusing. Consider using 'acquired' vs. 'non-acquired' or a more straightforward term.
- It would strengthen the manuscript to include a brief but explicit discussion on potential biases introduced by uneven geographical sampling intensity (e.g., higher representation from Europe/North America compared to Sub-Saharan Africa) and how this may affect global conclusions. The authors could clarify if seasonal variation or urban population size influenced ARG distributions, as these could be confounding variables.
- Additional context regarding public health implications or practical recommendations arising from the differential dispersal of acquired versus functional ARGs would be valuable for a broader readership.
- Consider integrating comparative genomic or literature-based evidence to further validate FG ARG annotations and discuss their potential to become mobilized.

(Remarks on code availability)

Reviewer #3

(Remarks to the Author)

in their manuscript entitled "geographics and bacterial networks shape the global urban sewage resistome", Martiny and colleagues describe the mining of metagenomic sequencing data from 1240 sewage samples for the presence of antimicrobial resistance genes. The authors distinguish "acquired" and "functional" antimicrobial resistance genes, and show that the effects of geography are distinct for both groups. Using a network analysis, the authors correlate microbial taxa and antimicrobial resistance genes.

While I think the sample set is impressive, there are several decisions in the analysis that seem unintuitive to me. First off, I am unsure what the biological basis for splitting the genes in "acquired" and functional is, and I think this point is insufficiently made clear. As I understand it, the "acquired" group are genes that have been experimentally shown to confer antimicrobial resistance (AMR), and also to be mobilized. On the other hand, the "functional" group represents a collection of genes shown to confer antimicrobial resistance by functional metagenomics (random high throughput cloning and heterologous expression in a host). It is unclear to me the way a gene was shown to confer AMR is relevant in grouping the genes for the analyses presented in the manuscript. I think the argument is that genes identified by functional metagenomics are inherently less mobile, but as far as I can tell this isn't shown. Why wasn't, for example, target compound class chosen as a basis for grouping the genes?

The database size for screening seems small to me. I understand that you chose to limit the analyses to experimentally proven AMR genes, but as you mention in the introduction (line 119), other studies have identified at least an order of magnitude more putative AMR genes. Why did you choose to limit the analyses here to the experimentally identified genes? This strikes me as a missed opportunity given the large and broad sample set. In addition, the numbers in line 313-315 suggests that ~85% of the reads mapping to AMR genes were not mapped to the categories analysed in the manuscript, why were these data not included?

The why was 25% prevalence chosen for the network analyses presented in figure 4? I would assume that this biases the analyses towards bacteria present in the human gut, as is seen in several of the network communities? The rationale and implication of this decision should be discussed in more detail.

Why was no metadata, other than sampling location, on the sewage samples included in the analyses?

The analyses in the manuscript are impressive, but I would like to see a more in depth discussion of the meaning of the results (eg the distance decay relations observed).

minor remarks:

The text in the figures (main text and supplement) is generally much too small

versions of tools should be given, and settings (eg of ARGfinder pipeline) should be described in more detail to help evaluate the results.

The numbers in the section "summary of sewage sample metagenomes" (starting line 297) don't seem to be consistent. line 303-304 state that 0.22% of the reads mapped to mOTUs and 0.16% to PanRes. That doesn't add up to 0.36% overall. More concerning, line 307 says there's 1394 million reads mapping to mOTUs, whereas line 313 says there's 103 million reads mapping to PanRes. Since that's over a 10 fold difference, how can the relative numbers be 0.22 and 0.16%?

line 571 I don't think it is true that bacteroidota and bacillota constitute over 90% of gut species, see eg the UHGG paper (<https://www.nature.com/articles/s41587-020-0603-3>)

(Remarks on code availability)

Version 1:

Reviewer comments:

Reviewer #1

(Remarks to the Author)

Authors should be commended for revising much of the analysis, code, and manuscript. The code is very well organized. There are still minor fixes to be made like adding model results to the scatter plots/regression lines so the figure is completely understandable on its own, but all-in-all the manuscript is much improved.

(Remarks on code availability)

Reviewer #2

(Remarks to the Author)

The revised manuscript presents an extensive global-scale analysis of antimicrobial resistance gene (ARG) distributions in sewage samples from various cities worldwide. The study distinguishes between ARGs identified through known mobilization pathways (acquired ARGs) and those identified solely via functional metagenomic (FG ARGs) methods. Based on a robust dataset of 1,240 samples from 351 cities across 111 countries, the authors provide evidence for differential global dispersal patterns and ecological associations of these two ARG classes.

KEY RESULTS:

- Demonstration of clear global and regional differences in the distribution of acquired versus FG ARGs, with acquired ARGs exhibiting stronger regional clustering.
- Identification of distinct distance-decay relationships, indicating that acquired ARG dispersal is more limited regionally, whereas FG ARGs show a more gradual global dispersion.
- Network analyses reveal stronger ecological associations between FG ARGs and specific bacterial communities, indicating a potential latent reservoir of ARGs that could pose future clinical risks.

SIGNIFICANCE TO THE FIELD:

This manuscript enhances our understanding of ARG dissemination by distinguishing between acquired and functionally identified ARGs, providing novel insights into the ecological and evolutionary mechanisms driving ARG distribution globally. While this study represents a continuation of prior publications by the Global Sewage Consortium (notably doi: 10.1038/s41467-019-08853-3 and 10.1038/s41467-022-34312-7), the current work provides substantial added value through its extensive dataset, extended analytical approaches (e.g., detailed network and distance-decay analyses), and explicit differentiation between ARG types.

DATA, METHODOLOGY, REPRODUCIBILITY

The scale and diversity of the dataset are commendable. The extensive dataset, sequencing, bioinformatic pipeline (ARGprofiler, metaSPAdes), and compositional statistical analyses (CLR and ALR transformations) are appropriate and

well-documented. Following the revisions, detailed methodological descriptions (including software versions and settings) are adequately provided, ensuring the work can be reproduced.

INTERPRETATION, CLAIMS, ADDITIONAL EVIDENCE:

The authors partially address initial reviewer concerns regarding the interpretation of FG ARGs, explicitly acknowledging that their functional role outside experimental conditions remains speculative. The revision improved the discussion on potential sampling biases and limitations arising from uneven geographic representation and seasonal variations.

RECOMMENDATIONS FOR REVISION:

Provide additional clarity regarding the implications of methodological biases (e.g., FG ARG detection associated with specific bacterial hosts).

While adequately justified, future work or follow-up studies should aim to experimentally validate the ecological and clinical relevance of FG ARGs.

Critically, the differentiation between 'acquired' and 'functional metagenomic' ARGs remains unclear and was flagged as problematic by all three reviewers. The central novelty and conclusions of this manuscript are based on this distinction, which I remain unconvinced is biologically meaningful or sufficiently justified.

(Remarks on code availability)

Reviewer #3

(Remarks to the Author)

The revised version of "Geographics and bacterial networks shape the global urban sewage resistome" by Martiny and colleagues reads well, and addresses many of the suggestions I and the other reviewers had. One thing that isn't fully clear to me yet is how the partitioning of the references in the PanRes database into "acquired", "functional", and a subset not included in the study affects the outcome of the analyses. In their first version, the authors indicate that the majority of fragments mapping to the PanRes reference database are not mapping to the chosen subsets of acquired and functional.

Please correct me if I'm interpreting this wrong: Ideally, acquired and functional represent biologically meaningful categories, where acquired represents genes acquired in response to antibiotic stress and functional the gene pool involved in microbial chemical communication and warfare that hasn't been mobilized in response to anthropogenic antibiotic stress (yet). If this interpretation is correct then the rest of the Panres database references should also fall in either of these categories, regardless of whether there is evidence for either yet. Since the mapping data is available, I think it would be appropriate to see if the effects observed in the manuscript are robust to the inclusion of this broader set.

I also came across a few minor points:

One of the communities is removed from figure 4. I may have missed this in the reviewer response, but What was the reasoning for removing the 7th community? Related, figure S10 still contains seven communities.

There is a small mistake in the legend of fig 2 & 3, two of the trend lines are referred to as "purple".

This might be a rendering issue on my end, but several of the plots in the additional data file are missing.

(Remarks on code availability)

Version 2:

Reviewer comments:

Reviewer #2

(Remarks to the Author)

I want to thank the authors for their thorough and constructive revisions. The additional analyses and clarifications have strengthened the manuscript and convincingly addressed the concerns raised in the previous review round.

Overall, this is an important contribution that will be of broad interest to the microbial genomics and public health

communities. I am pleased with the revised version.

(Remarks on code availability)

Reviewer #3

(Remarks to the Author)

In the third round of review, Martiny and colleagues have addressed the concerns me and other reviewers had by extending their analyses to the "Other ARGs" that weren't assigned to the categories in the main manuscript. This extra analysis is insightful, but currently only made available to the reviewers.

Thank you for so thoroughly addressing the reviewer remarks. I would suggest that these analyses are included in the supplementary information of the paper, so that they are also available to readers who may have the same questions as the reviewers did.

(Remarks on code availability)

We appreciate the time the reviewers took to comment on our manuscript, and we have incorporated several of their suggestions. The changes in the manuscript related to comments by reviewer 1 are highlighted in **red**, by reviewer 2 in **blue**, and by reviewer 3 in **green**. Below is our point-by-point response to the reviewers' comments in *italics*.

We have addressed the concerns from reviewer #1 regarding the analysis of differences in slopes in our distance-decay models by testing whether they are different using the emmeans package in R, and have thus rewritten that part of our results section. We have also revised our discussion to address the concerns raised by reviewer #2 regarding the validation of the functional ARGs, potential biases, and confounding variables in our study. Other changes to the manuscript include a rewritten introduction to address unclear points, version numbers added to programs in methods, and other minor fixes.

Reviewer #1

These are reviews for the paper by Martiny and colleagues entitled "Geographics and bacterial networks shape the global urban sewage resistome" (#NCOMMS-25-08182). The paper wields a large dataset to tussle with some huge important questions about the biogeography of antibiotic resistance genes and is remarkably resemblant of another distance-decay paper by an author of the same surname.

This is correct that we applied the same approach of analysing distance-decays outlined in Martiny et al. (2011, PMID 21518859) as cited in our methods section (line 262-263). Both papers find that there is a distance-decay effect. However, Martiny et al. (2011) data are based on 16S profiling of soil samples, while our work focuses on antimicrobial resistance genes in metagenomic sewage data. Further, the effects (slope, etc.) are not similar. There is also a family relation, but we are not working together.

While the dataset is impressively large spanning >300 cities and the paper seems mostly methodologically sound, it is difficult to read, the distance-decay statistical models are incomplete, and presentation of the results are piecemeal or hurried.

The specific comments below are intended to help improve the manuscript.

I do like the overall structure of the results: summary of the metagenomes, geographic distribution, distance-decay, network analysis.

L80-81 Not clear on what a 'large number' is. About how many? Same for 'considerably larger number' on L85. Same for 'large amount' and 'tiny fraction' on L113.

Our introduction has been revised to enhance clarity, and the sentences with "large number" and "considerably larger number" have been removed from our manuscript. The sentence in L113 in the original manuscript has also been moved and modified to specify that it's only 5-15% of all ARGs being studied in lines 82-85.

The second paragraph of the introduction does a poor job of setting up the problem and the study.

We have rewritten the introduction to enhance its clarity, where we setup the problem and the study in lines 80-95.

The factors at play (regional [cultural, economic, etc.], distance, and bacterial (host) communities) are not disentangled clearly for the reader by the end of the paragraph.

This has been changed to be in a paragraph about epidemiological tracking (lines 97-109), and the following paragraph about distance-decays (lines 111-122), which should clarify our points.

The fourth paragraph could be trimmed of distracting points on the different databases to better lead to the hypothesis.

We have shortened this section and refer simply to the two classes of databases: those relying on functional selections, and those that do not - see lines 124-126.

The hypothesis is also not quite clear because human-associated bacteria (L26) are also bacterial taxa (L25). Please clarify the terminology.

We guess the reviewer refers to L125 and L126 in the first version of the manuscript, and agree that the original sentence was unclear. We have revised the sentence to explicitly distinguish between environmental bacterial taxa and those associated with human gut (lines 129-130).

I assume the genes in ResFinder are assumed 'acquired' because they were most likely from genomes of isolated pathogens. And, because they are pathogenic, and I assume actively infecting a patient (maybe taking antibiotics) at the time they were isolated, they experienced selection pressure to 'acquire' antimicrobial resistance. This is different from metagenomics which targets mostly non-pathogenic bacteria. Therefore the selection pressure to acquire antimicrobial resistance is much lower. More likely is that these ARGs play a non-pathogenic 'functional' role, possibly even not related to antimicrobials at all. Therefore, these genes are more controlled by environmental conditions suitable for growth of the host (pH, temperature, oxygen, osmolarity, nutrient availability, etc.) akin to Baas Becking's hypothesis. It follows that human associated (opportunistic) pathogens might disperse more widely by hitching rides with humans. I don't want to rewrite the introduction, but the authors have to straighten this out.

We can see how our meaning behind the word was not entirely clear and have now defined it more clearly (lines 80-95). They have been 'acquired' naturally by some members of a species, meaning they are accessory genes (rather than core) to that (yes, often a pathogenic) species. These acquired ARGs are not only found in pathogens, but are widespread in commensal bacteria, where they are naturally selected for, for example, the gut microbes are exposed to antibiotics.

The functional genes have so far not been identified as naturally acquired, but they were

shown to provide resistance if acquired in a cloning experiment. The reviewer is correct that these might not provide resistance functionality in the context we are observing them in this study; however, we argue that these FG ARGs may represent a reservoir of resistance that could be mobilized in the future.

L133-134 This seems to 'confirm' the opposite of the hypothesis already. Please clarify.

We have rewritten the sentence in lines 136-139 to clarify that we saw the opposite of our original hypothesis.

L134 Remove 'highly'.

The word 'highly' has been replaced with 'relatively' to accommodate the next point by the reviewer (line 139).

L135 You'll want to make this clear that this is relative abundance (vs. absolute abundance)

We agree. It has been implemented in line 139.

L152 "spun down" is informal. Also, what g force and for how long?

We have changed the informal writing to include g force and for how long in lines 155-156, but maintain the reference to the full protocol in the paper by Knudsen et al. (2016) (PMID: 27822556).

L308-309 'named' vs 'unnamed' may be a little vague. Maybe 'latinized' vs 'placeholder' would be more specific and has been used in one other case I know of:

<https://gtdb.ecogenomic.org/stats/r220>

That is a good suggestion and we have changed the 'named' to be 'latinized' and unnamed to be 'placeholder' in both the methods section (line 200) and in the mentioned result sentence in lines 323-324.

L340 Did the PCA and permanova explain the same amount of variation? What was the permanova's R-squared value?

No, for the PCA, we only provide the variance explained by the first two principal components as seen in figure 1b and 1c. For the permutational ANOVA test, we calculate the proportion of variation explained by world region, which is simply the R2 value multiplied by a 100.

L343 Procrustes analysis compares two arrangements in reduced space, where the arrangements are always imperfect representations of the calculated dissimilarities. Wouldn't it be better to test the differences of the dissimilarities directly with something like a mantel test?

Excellent suggestion by the reviewer, where we have now added the Mantel test results for comparison of dissimilarity between bacterial abundances and acquired and FG abundances, see lines 390-392.

L362 Many decay rates are referred to, but only one is reported. What test are the p-values from? The stats in this section need some work. Could a more sophisticated model, just adding an interaction term (within or between different countries), maybe, be more direct in asking about differences between slopes? If not, maybe confidence intervals (with `confint()` maybe) can be used to gauge differences in slope? More description in the Table S2 caption would also really help.

These are some excellent suggestions, and we have updated our distance-decay analyses to include a test of interactions between the slopes, confirming whether there is a difference between the different spatial scales (new Table S3). We've done a pairwise comparison of the slopes of each DD model with the R package emmeans. We have rerun the entire analysis and updated the results, including Tables S2 and S4 (as per the next two comments from the reviewer).

Table S2 Some symbols are not reproducing in the table leading to ambiguity in what the authors mean. Mantel tests are usually done with two dissimilarity matrices. Which two are being compared in each row? Plus, these aren't summary stats. They are results of hypothesis testing.

We apologize to the reviewers for the incorrect reproduction of the results in Tables S1 and S2. We have updated the formatting of both tables and rerun our analyses to correct the presented results (lines 374-437). Furthermore, we have updated the descriptions for the information in Table S1 and S2.

Table S3 More description of these data are necessary.

We have updated the data description in Table S3; see the comments above.

L369 It may have affected it, but more specifically, you didn't detect any effect.

We have updated our wording to reflect that we did not detect any effects in line 386.

Fig2abc & Fig3cd It is difficult to see the pink/orange circles compared to the other points. For completeness, it would be great to have in the caption or somewhere locally the p-values for the regression lines, but also to know if the slopes significantly differed.

We have changed the order in which each colored marker is plotted in those figures to make the smaller sample groups more visible. We have not included the p-values in the figures, but refer the reviewer to Table S2.

Fig 4 These figures are very busy and could benefit from some reduction and/or some visual clarification (e.g., grouping). The legend is too small in Fig4a to make much use of.

We thank the reviewer for the suggestion. In the revised version of Figure 4, we improved the clarity and readability of the network by increasing the visual separation between communities. This was achieved by removing peripheral nodes with low connectivity and by improving the layout tuning to better highlight the modular structure. Additionally, we enlarged the legend in panel 4a to make it more legible.

Table S1 The R-squared values are all pretty low, despite significance.

We agree with the reviewer that the R-squared values are quite low and have revised our description of the results in Table S1 (lines 360-362).

L362,364,369,533 From what I can tell, even after reviewing the code, differences in slope were never directly tested by including an interaction term, so these statements aren't justified. What is function `fit_dd_model2()` do?

The fit_dd_model2 was just a wrapper function, but we've cleaned up the code functions now and added a test of interaction terms. Our GitHub has been updated to reflect the changes in our code.

Reviewer #2

This manuscript by Martiny, Munk, Fuschi, et al., as part of the Global Sewage Consortium, investigates the global distribution patterns of antimicrobial resistance genes (ARGs) in urban sewage, distinguishing between ARGs known to be mobilized (acquired ARGs) and non-mobilized genes identified via functional metagenomics (FG ARGs). The authors leverage an extensive dataset comprising metagenomic sequencing data from 1,240 samples collected from 351 cities across 111 countries between 2016 and 2022. The study employs robust bioinformatic pipelines, statistical modeling, and network analyses to elucidate the underlying ecological and geographical factors influencing ARG distribution.

The authors should be commended for undertaking this massive collaborative global effort, which has already provided key insights into the role of wastewater as a powerful tool for antimicrobial resistance (AMR) surveillance worldwide.

However, to this reviewer, the results in this manuscript represent a continuation of two previous publications from this same consortium (doi: 10.1038/s41467-019-08853-3 and 10.1038/s41467-022-34312-7) in describing the global distribution of AMR genes, albeit with an expanded sample set and geographical coverage. The novelty in this work primarily lies in the network analysis and the differentiation between acquired and functional metagenomic resistance genes. This analysis highlights distinct dissemination patterns between these two classes and demonstrates that the resistome in sewage is largely shaped by its microbial composition. While confirming that this pattern applies globally is interesting, previous studies have already shown similar microbiome-driven resistome structuring in human and wastewater samples (e.g., doi: 10.1038/nature17672).

Key results: The authors report significant differences between acquired and FG ARGs in their global distribution and abundance. The acquired resistome displayed strong

geographical structuring, closely following regional distinctions, whereas the FG resistome showed more even global distribution. Additionally, clear distance-decay effects were demonstrated, with acquired ARGs primarily showing decay at national and regional scales, whereas FG ARGs exhibited global-scale dispersal limitations. Network analysis provided evidence that FG ARGs have stronger associations with bacterial taxa than acquired ARGs, suggesting different mechanisms of transmission and evolutionary pressures.

Validity; The data presented are technically sound, benefiting from a robust and well-documented methodology. Sequencing and analytical pipelines, such as ARGprofiler, metaSPAdes assemblies, and statistical analyses using compositional methods and network analyses, are appropriate and widely used in the field. Using compositional data analysis (CLR and ALR transformations) strengthens the validity of the findings. However, further validation of the functional ARG annotations, possibly by additional experimental or comparative genomic methods, would be beneficial.

Significance: The study advances our understanding of ARG dissemination at a global scale, clearly differentiating between ARGs based on their mobilization histories. These insights are critical for informing global surveillance efforts and policies to mitigate antimicrobial resistance.

Data and methods: The extensive global dataset and robust bioinformatics pipeline enhance the quality and reliability of the findings. Supplementary materials adequately complement the main findings, but the supplementary data files could benefit from clearer metadata annotations or indexing to facilitate independent verification and reuse.

Clarity and context: Overall, the manuscript is clearly written and well-organized. The findings are appropriately contextualized within existing literature. However, the authors could improve clarity by summarizing key methodological steps at the beginning of the results section to guide readers through complex analyses. Additionally, certain figures (e.g., PCA plots and UMAP clustering) could be visually optimized for better readability, especially given the extensive geographical data.

We have updated the Biplots in Figure 2 and UMAP plots in Figure 3 for improved readability.

Suggested improvements:

- The terms 'acquired' and 'functional metagenomic' for the two classes of genes are somewhat confusing. Consider using 'acquired' vs. 'non-acquired' or a more straightforward term.

We agree that the two terms can be confusing to the reviewer and other readers. However, we do not think that naming the FG genes as non-acquired is justified, as there is no evidence to confirm whether they can be acquired or not. Since it was not within the scope of this article to investigate whether the FG ARGs had already been mobilized, we decided to group those ARGs in a category reflecting how they were identified. The 'acquired' term for the ResFinder genes is more accurate, since the ResFinder database only includes mobilized/acquired ARGs.

- It would strengthen the manuscript to include a brief but explicit discussion on potential biases introduced by uneven geographical sampling intensity (e.g., higher representation from Europe/North America compared to Sub-Saharan Africa) and how this may affect global conclusions.

Indeed, the reviewer was correct to comment our lack of discussion of biases. We have included comments on the uneven geographical sampling in lines 558-560 and seasonality in lines 592-594.

The authors could clarify if seasonal variation or urban population size influenced ARG distributions, as these could be confounding variables.

The reviewer's suggestion is good; therefore, we have added a sentence to the discussion about seasonal variation and urban population size (lines 592-594). We would, however, like to note that we did not identify these as important variables previously when analysing data from sewage sampled up until 2019 (Njage et al. 2023, PMID: 37940209).

- Additional context regarding public health implications or practical recommendations arising from the differential dispersal of acquired versus functional ARGs would be valuable for a broader readership.

Another good suggestion, where we have added in two places in the discussions about the implications of distance-decay trends in lines 579-581 and implications for AMR surveillance 605-608.

- Consider integrating comparative genomic or literature-based evidence to further validate FG ARG annotations and discuss their potential to become mobilized.

We agree that further validation of the functional ARGs would add significant value to our analysis; however, it is outside the scope of our paper, which focuses on studying the differences in distribution, abundance, and bacterial contexts of the acquired and FG ARGs. The paper by Daruka et al. (2025) included experimental validation of their genes, and the ResFinderFG collection was constructed through a literature search of published studies that reported new ARGs.

Reviewer #3

in their manuscript entitled "geographics and bacterial networks shape the global urban sewage resistome", Martiny and colleagues describe the mining of metagenomic sequencing data from 1240 sewage samples for the presence of antimicrobial resistance genes. The authors distinguish "acquired" and "functional" antimicrobial resistance genes, and show that the effects of geography are distinct for both groups. Using a network analysis, the authors correlate microbial taxa and antimicrobial resistance genes.

While I think the sample set is impressive, there are several decisions in the analysis that seem unintuitive to me. First off, I am unsure what the biological basis for splitting the genes in "acquired" and functional is, and I think this point is insufficiently made clear. As I understand it, the "acquired" group are genes that have been experimentally shown to confer antimicrobial resistance (AMR), and also to be mobilized. On the other hand, the "functional" group represents a collection of genes shown to confer antimicrobial resistance by functional metagenomics (random high throughput cloning and heterologous expression in a host). It is unclear to me the way a gene was shown to confer AMR is relevant in grouping the genes for the analyses presented in the manuscript. I think the argument is that genes identified by functional metagenomics are inherently less mobile, but as far as I can tell this isn't shown. Why wasn't, for example, target compound class chosen as a basis for grouping the genes?

We agree that the rationale for dividing the ARGs into "acquired" and "functionally identified" (FG) groups were not clearly written in our first submission of the manuscript, which we have revised to better explain the distinction (lines 92-97) and relevance to our study (lines 127-133). We chose to split the ARGs into those two groups to reflect their ecological and evolutionary contexts, as the acquired ARGs likely represent the current burden of resistance and the FG ARGs to represent a latent reservoir of resistance that may be at risk of becoming mobilized in the future. The reviewer is correct in that we have not shown directly if the FG ARGs are less mobile, but we believe our distance-decay models support this idea.

Regarding whether to group the genes by their target compound class, we agree that this is of biological relevance and has also presented results on the abundances at class levels (for example, Table S1).

The database size for screening seems small to me. I understand that you chose to limit the analyses to experimentally proven AMR genes, but as you mention in the introduction (line 119), other studies have identified at least an order of magnitude more putative AMR genes. Why did you choose to limit the analyses here to the experimentally identified genes? This strikes me as a missed opportunity given the large and broad sample set.

We appreciate the suggestion of including the putative ARGs identified by Inda-Diaz, but we chose to focus on those ARGs that have been experimentally validated to ensure as high confidence in the phenotypes and maintain comparability with our previous sewage studies (PMID: 30850636 , PMID: 36456547).

While the predicted ARGs from Inda-Diaz et al. (PMID: 36882798) offers a broader coverage, they were restricted to well-studied resistance classes. In contrast, the ARGs from Daruka et al. (PMID: 39805953) include genes to novel antibiotics, which were inline with our goal of studying emerging threats of resistance. However, we agree that for future work it would be very interesting to build upon the work of Inda-Diaz et al.

In addition, the numbers in line 313-315 suggests that ~85% of the reads mapping to AMR genes were not mapped to the categories analysed in the manuscript, why were these data not included?

We have removed these numbers, as they were not relevant to our study of the sewage resistome differences between acquired and FG ARGs.

The why was 25% prevalence chosen for the network analyses presented in figure 4? I would assume that this biases the analyses towards bacteria present in the human gut, as is seen in several of the network communities? The rationale and implication of this decision should be discussed in more detail.

We acknowledge that a few more words are warranted. As the text indicates, the 25% prevalence was chosen "To increase the signal-to-noise ratio and reduce sparseness" (lines 276-278). We now clarify that high sparseness leads to many replaced values that might end up skewing the results. We are not aware of any automated way of picking this threshold: it is always a balancing act between sensitivity and robustness.

Why was no metadata, other than sampling location, on the sewage samples included in the analyses?

We appreciate the reviewer's interest in analyzing additional metadata, but unfortunately, a lot of the other metadata was of low quality, which is why we limited our analyses to the geographical locations to study the differences in dispersal of the two ARG groups. However, we do agree that integrating richer metadata would be important in future work.

The analyses in the manuscript are impressive, but I would like to see a more in depth discussion of the meaning of the results (eg the distance decay relations observed).

We have rewritten our discussion to provide a more in-depth discussion of our results in the revised manuscript: distance-decay in lines 554-564, network results in lines 566-581 and the relevance of including FG ARGs in lines 583-592.

minor remarks:

The text in the figures (main text and supplement) is generally much too small

We have increased the font size in some of the examples that we agree have too small font sizes. We have prioritized the main text, which some people may choose to print on paper.

versions of tools should be given, and settings (eg of ARGfinder pipeline) should be described in more detail to help evaluate the results.

Indeed, the first version of our manuscript did not include versions and settings for some of the tools used, but we have updated the corresponding sections with these details. See lines 167-181, 199, 241, 300.

The numbers in the section "summary of sewage sample metagenomes" (starting line 297) don't seem to be consistent. line 303-304 state that 0.22% of the reads mapped to mOTUs and 0.16% to PanRes. That doesn't add up to 0.36% overall. More concerning, line 307 says there's 1394 million reads mapping to mOTUs, whereas line 313 says there's 103 million

reads mapping to PanRes. Since that's over a 10 fold difference, how can the relative numbers be 0.22 and 0.16%?

We have double-checked the numbers, and the reviewer was correct in spotting these mistakes. We have corrected the numbers and rewritten that result section with the correct numbers; see lines 314-319, 321-322, 328-333.

line 571 I don't think it is true that bacteroidota and bacillota constitute over 90% of gut species, see eg the UHGG paper (<https://www.nature.com/articles/s41587-020-0603-3>)

We agree that this sentence needs some refinement. These two phyla comprise 76% of the referenced UHGG collection. However, here, we are referring to their proportion (abundance) in the gut microbiota composition of healthy individuals. We have clarified this point in the manuscript in line 572.

We appreciate the time the reviewers took to comment on our manuscript, and we have incorporated several of their suggestions. The changes in the manuscript related to comments by reviewer 1 are highlighted in **red**, by reviewer 2 in **blue**, and by reviewer 3 in **green**. Below is our point-by-point response to the reviewers' comments in *italics*.

Other updates to the manuscript include a new title and minor adjustments to the main text. We have added material for reviewers 2 and 3 containing our re-analysis of the excluded PanRes genes, which is located after the individual response to each reviewer comment in this document.

Reviewer #1

Authors should be commended for revising much of the analysis, code, and manuscript. The code is very well organized. There are still minor fixes to be made like adding model results to the scatter plots/regression lines, so the figure is completely understandable on its own, but all-in-all the manuscript is much improved.

We have updated Figures 2, 3, and S9 to include the model results, as per the reviewer's suggestion.

Reviewer #2

RECOMMENDATIONS FOR REVISION:

Provide additional clarity regarding the implications of methodological biases (e.g., FG ARG detection associated with specific bacterial hosts).

We agree that there are many things to discuss about the methodological biases about FG ARGs, as we highlighted in lines 586-595.

While adequately justified, future work or follow-up studies should aim to experimentally validate the ecological and clinical relevance of FG ARGs.

We agree with the reviewer's comments about experimental validation of the FG ARGs, as discussed in our manuscript (lines 586-602). We added an additional comment on their clinical relevance in line 602.

Critically, the differentiation between 'acquired' and 'functional metagenomic' ARGs remains unclear and was flagged as problematic by all three reviewers. The central novelty and conclusions of this manuscript are based on this distinction, which I remain unconvinced is biologically meaningful or sufficiently justified.

We appreciate the reviewer's concerns and have chosen to compile additional material for reviewers #2 and #3 to justify our categorization of the ARGs. Specifically, we have provided a detailed overview of the PanRes database and the rationale for only working on a subset

of the ARGs. We acknowledge that the distinction between acquired and FG ARGs is not absolute, but our analyses consistently show distinct ecological patterns, which we believe justifies our analytical separation into biologically relevant categories. This material can be found in this document after our responses to the reviewers' comments.

Reviewer #3

The revised version of "Geographics and bacterial networks shape the global urban sewage resistome" by Martiny and colleagues reads well, and addresses many of the suggestions I and the other reviewers had. One thing that isn't fully clear to me yet is how the partitioning of the references in the PanRes database into "acquired", "functional", and a subset not included in the study affects the outcome of the analyses. In their first version, the authors indicate that the majority of fragments mapping to the PanRes reference database are not mapping to the chosen subsets of acquired and functional.

Indeed, the reviewer is correct in noticing that we removed the results about mapping statistics to the complete PanRes database. Upon reading the reviewer comments and reviewing this choice, we have decided to compile additional material for the reviewers, comparing the distributions of the acquired, FG, and excluded ARGs in the sewage resistomes. This material can be found in this document after our responses to the reviewers' comments.

In the material, we demonstrate how the excluded ARGs exhibit a mixture of characteristics observed in the acquired and FG ARGs, indicating that they contain ARGs that are both mobilized and non-mobilized. However, due to a lack of explicit annotations on their origin and mobility, these excluded genes were not included in the main manuscript.

Based on the work presented in this manuscript, we acknowledge that a comprehensive evaluation of the strengths and weaknesses of the different ARG reference databases is necessary. This, however, falls outside the scope of this paper, which is why we have removed the part of the PanRes mapping statistics from the main text. But it would be an exciting topic of future work.

Please correct me if I'm interpreting this wrong: Ideally, acquired and functional represent biologically meaningful categories, where acquired represents genes acquired in response to antibiotic stress and functional the gene pool involved in microbial chemical communication and warfare that hasn't been mobilized in response to anthropogenic antibiotic stress (yet). If this interpretation is correct then the rest of the Panres database references should also fall in either of these categories, regardless of whether there is evidence for either yet. Since the mapping data is available, I think it would be appropriate to see if the effects observed in the manuscript are robust to the inclusion of this broader set.

We appreciate the reviewer's suggestion of examining whether our observed effects of the resistance genes alter our conclusions by including the excluded other PanRes ARGs. As explained in the previous comment, we have compiled additional material showing how these other ARGs behave and this material can be found at the end of this document after our reviewer responses. The reviewer is correct in their interpretation of our categories of

acquired and FG ARGs. However, we lack clear annotations on the mobility and discovery methods of these other ARGs, which is why they were not included in the main manuscript.

I also came across a few minor points:

* One of the communities is removed from figure 4. I may have missed this in the reviewer response, but What was the reasoning for removing the 7th community? Related, figure S10 still contains seven communities.

We thank the reviewer for bringing this inconsistency to our attention. As also suggested by another reviewer, we revised Figure 4 to provide a cleaner and more readable representation by removing peripheral nodes from all communities. In this process, community 7 was excluded due to its small size and because it was not further discussed in the manuscript.

We have also updated Figure S10 to exclude communities 7 and 8, ensuring consistency with the main text and figures. We apologize for the oversight in the previous version.

There is a small mistake in the legend of fig 2 & 3, two of the trend lines are referred to as "purple".

Thank you to the reviewer for bringing this to our attention. We have updated the legends in Figures 2 and 3, as well as Figure 1, following the identification of another error in the figure text.

This might be a rendering issue on my end, but several of the plots in the additional data file are missing.

We apologize for any rendering issues and have double-checked our plots, submitting them as individual files.

Supplementary material for reviewers

We have compiled this supplementary material to address our decision to focus solely on a specific subset of the PanRes database, specifically the genes from the databases ResFinder¹, ResFinderFG², and the CsabaPal³ collections.

In this material, we begin by explaining the structure of the PanRes collection and the reasons for dividing a subset of the ARGs into acquired and functional metagenomic (FG) genes. Next, we expand the primary analyses of the manuscript by including the other excluded ARGs, which we refer to as the “other ARGs”. Finally, we discuss how the exclusion of specific genes may have affected our main conclusions.

The content of PanRes

We introduced the idea of PanRes in our ARGprofiler manuscript⁴ and made the full collection publicly available on Zenodo (<https://zenodo.org/records/13885013>). In brief, the PanRes collection integrates multiple existing resistance gene reference databases into a single, non-redundant resource, thereby eliminating the need to select which resistance reference database to use. Specifically, we included genes from the sources listed in Table 1.

As shown in Table 1, only the three databases ResFinder, ResFinderFG, and CsabaPal do not contain ARGs identified through similarity predictions. While overlaps exist across all the different database sources (Figure 2), the functional metagenomic collections are mostly unique (Figure 3).

Our study aimed to investigate the latent reservoir of resistance genes in sewage resistomes, which may represent a collection of ARGs that may become mobilized in the future. To achieve this, we focused on ARGs known to be mobilized or acquired, as well as those identified through functional metagenomics, which we hypothesize to be intrinsic resistance genes. We grouped the ARGs into these categories:

- Acquired ARGs: Genes known to be mobilized and transferred between species. We used ResFinder genes for this group, as this collection is explicitly curated for acquired resistance⁵.
- FG ARGs: ARGs identified with functional metagenomics (FG), which are believed to be intrinsic to environmental bacteria, and not yet mobilized. The ResFinderFG and CsabaPal collections contain solely genes identified with this technique.
- Other ARGs: Genes from the remaining databases that do not clearly fall into the other two categories. These are likely to include a mix of both types; however, due to a lack of annotations regarding mobilization and discovery, they were excluded from the main study.

Database	Version	# ARGs	Contains ARGs	Contains similarity-predicted genes	Identified with functional metagenomics
----------	---------	--------	---------------	-------------------------------------	---

ResFinder ¹	2023-01-20	3,131	Yes	N	No**
ResFinderFG ²	2.0	3,897	Yes	N	Yes
CARD ⁶	3.2.5	4,661	Yes	Yes	No**
MegaRes ⁷	3.0.0	8,295	Yes	Yes	No**
AMRFinderPlus ⁸	3.11/2022-12-19.1	6,465	Yes	Yes	No**
ARG-ANNOT ⁹	V6_July2019	2,223	Yes	Yes	No**
CsabaPal ³	November 2022	1,093	Yes	No	Yes
MetalResistance*	https://doi.org/10.5281/zenodo.8108201	578	No	No	No

Table 1: **The databases included in PanRes v1.0.1**, including their version number, how many reference sequences were included, whether they contain ARGs or not, and an indicator of whether the source contains genes identified through similarity predictions.

* *MetalResistance* is an in-house collection that contains both genes from BacMet¹⁰ v1.1 and a manual query.

** Since the ResFinderFG genes were identified by reviewing published studies, several genes are already included in the other databases.

98% cluster database overlaps

Figure 1: **Upset plot showing the overlap for all PanRes databases in their representative members for the 98% CD-HIT clustering.** Highlighted rows show the chosen databases for which we created groupings: ResFinder, ResFinderFG, and CsabaPal. The coloring follows the scheme used in the main paper to distinguish between Acquired (red) and FG (blue) ARGs.

Figure 2: **Overlap in representative cluster sequences between the groupings** of Acquired (ResFinder), FG (ResFinderFG+CsabaPal), and the other ARGs not included in the main manuscript (CARD, MegaRes, AMRFinderPlus, ARGANNOT).

Primary analyses

To determine whether our focus on acquired and FG ARGs introduced any bias into our study, we have compared the abundance, diversity, and variants of all three ARG categories.

Read abundances

To start with, we compared the number of sequences matched with at least one read fragment, the total number of fragments aligned, and the sample-wise summary statistics (average, range, and standard deviations) of fragment counts, as shown in Table 2. Among the representative sequences clustered at 98% identity, we observed that the acquired ARGs had the highest proportion of sequences that were hit. In terms of sample-wise fragment counts, FG and Other ARGs had similar levels, while the acquired ARGs were lower in read fragments aligned.

	Acquired	FG	Other ARGs	All ARGs
Representative 98% sequences hit (%)	1,052 / 1,119 (94.01%)	3,108 / 3,421 (90.85%)	1,512 / 1,672 (90.43%)	5,645 / 6,212 (90.87%)
Total fragments aligned to sequences (M: million)	17.28 M	21.75 M	31.01 M	70.05 M
Average of fragments aligned to sequences (M: million)	0.015 M	0.018 M	0.027 M	0.061 M
Range of fragments aligned (M: million)	0.0002 M - 0.113 M	0.0006 M - 0.239 M	0.001 M - 0.209 M	0.003 M - 0.438 M

Standard deviation of fragments aligned	0.013 M	0.016 M	0.019 M	0.043 M

Table 2: Summary of fragments aligned to the references in the three different groupings: Acquired, FG, and Other ARGs.

Next, we calculated the alpha- and beta-diversities of the fragment counts for the three different groupings (Figures 3 and 4). Figure 3 compares the aligned fragment of ARGs to the amount aligned to bacterial genera, the absolute richness, and the Shannon diversities. While roughly the same total amount of fragments were aligned, the FG and other ARGs exhibited higher richness and Shannon diversity than the acquired ARGs. Interestingly, the sample-wise diversity metrics of the other ARGs consistently fell between those of the FG and acquired ARGs, supporting our interpretation that this group likely represents a mixture of both mobilized and intrinsic resistance genes.

Continuing this, we saw that the country-wise abundances of the ARGs in Figure 4a also indicated that the other ARGs are a group of mobilized and intrinsic resistance genes, although the abundances were almost the same in some countries, for example, in Georgia, Tanzania, and Vietnam. For the beta diversity visualized in the PCA biplots in Figure 4, we saw that the beta diversity could be explained by 8.9% by world regions, which was slightly higher than the FG ARGs at 7.4% and lower than the acquired at 12% (Table 3). Similarly, the other ARGs were more related to the bacteriome than the FG ARGs and less than the acquired ARGs.

	Acquired ARGs	FG ARGs	Other ARGs
Beta diversity explained by regions (permanova)	12% (p = 0.001)	7.4% (p = 0.001)	8.9 % (p = 0.001)
Relatedness to bacteriome (procrustes)	0.88 (p=0.001)	0.69 (p=0.001)	0.79 (p=0.001)

Table 3: Permanova and Procrustes tests on the beta-diversity of the resistomes in relation to world regions and bacteriome.

Figure 3: Alpha-diversity indices of antimicrobial resistance genes (x-axis) and the bacterial genera (y-axis) found in the sewage samples, stratified by ARG grouping. **a.** The number of read fragments aligned to either the Acquired ARGs, FG ARGs, or Other ARGs against the fragments aligned to the bacterial genera. **b.** Absolute Richness of the ARGs and genera. **c.** Shannon diversity indices for ARGs and bacterial genera.

Figure 4: Abundance and beta-diversity of ARGs across geographical regions. **a.** The country-wise ALR abundances of acquired ARGs, FG ARGs, and Other ARGs. **b.** PCA biplots of resistance genes (98% homology grouping), in which the PCA loadings were calculated from CLR values, with the acquired ARGs left, FG ARGs in the middle, and the other ARGs on the right. Each marker represents a sewage sample and is colored by the world region.

Distance-decay effects

Building on the observed diversity patterns, we next investigated whether geographic distance between sampling sites contributes to differences in other ARGs beyond those observed in the main manuscript (Figures 5 and 6).

In the distance-decay models based on abundance data (Figure 5), the slopes for the other ARGs closely resembled those of the FG ARGs (Table 4). However, at the inter-regional scale, the slope for the other ARGs became slightly positive (slope_{between region}=0.012), more similar to the slope observed for the bacterial abundance (slope_{between region}=0.015). This suggests that, like the FG ARGs, the other ARGs may be more environmentally embedded and less constrained by geographic distance. Mantel tests further supported this, showing that the other ARGs were slightly less correlated to the bacteriome ($\rho=0.73$; $P=0.001$) than FG ARGs ($\rho=0.76$; $P=0.001$), but more so than the acquired ARGs ($\rho=0.7$; $P=0.001$).

When analyzing the variants of the other ARGs discovered in the metagenomic assemblies, we did not observe a strong clustering of cities sharing the same variants in the UMAP plot (Figure 6c). The distance-decay slopes for the other ARG were also weak across all spatial scales, indicating limited effects of geographic distance on the distribution of variants (Figure 6, Table 4). Moreover, there were no strongly significant differences in the distance-decay slopes (Table 5), further suggesting that the other ARGs are not facing strong dispersal limitations.

Figure 5: Distance-Decay relationships for the bacteriome and resistome city communities ($n=60,030$ pairwise comparisons) for the abundance of **a.** Acquired ARGs, **b.** FG ARGs, **c.** Other ARGs, and **d.** bacterial genera. The x-axis shows the pairwise city distances in kilometers (km), and the y-axis shows similarities. The dashed line represents the fit across all spatial scales, where the solid lines denote regressions fitted individually by different city-wise comparisons: national (red), cities within the same region (blue), and inter-regional (between regions, purple). Model parameters and adjusted R^2 values are listed for each model in its corresponding plot.

a. Overlap of assembled acquired variants**b. Overlap of assembled FG variants****c. Overlap of assembled Other ARG variants****d. Distance-Decay of assembled acquired variants****e. Distance-Decay of assembled FG variants****f. Distance-Decay of assembled Other ARG variants**
Spatial Scale • Within Country • Within Region • Between Regions

Figure 6: UMAP and Distance-Decay Analysis of resistomes. **a.-c.** UMAP clustering of shared variants among the cities for a. acquired ARGs, b. FG ARGs and c. Other ARGs. Only cities with more than 100 non-singleton alleles were retained, and Hellinger transformed and clustered with the UMAP algorithm.

Each marker represents a city, colored by region and sized according to the number of unique variants in that city. City labels were optimized using the ggrepel package to minimize overlap. **d.-f.** Distance-decay relationships for assembled city resistomes across different spatial scales for **d.** acquired (n=4,656 pairwise comparisons), the FG variants (n=16,471 pairwise comparisons), and **f.** Other ARGs (n=39,903 pairwise comparisons). The x-axis shows the pairwise city distances in kilometers (km), and the y-axis shows resistome similarities. The dashed line represents the fit across all spatial scales, where the solid lines denote regressions fitted individually for three spatial scales: cities within the same country (red), cities within the same region (blue), and cities that are in different regions (between regions, purple). Model parameters and the adjusted R^2 values are written for each model in its corresponding plot.

Model		Abundance				Assembly variants		
		Acquired	FG	Other ARGs	Genera	Acquired	FG	Other ARGs
Scale Independent	R ²	0.04	0.012	0.017	0.0065	0.27	0.089	0.034
	Slope	-0.036*** (P<<0)	-0.02*** (P=3.8e-155)	-0.025*** (P=1.5e-219)	-0.017*** (P=9.5e-88)	-0.12*** (P=3.01e-322)	-0.071*** (P=0)	-0.055*** (P=1.6e-299)
	Mantels r	0.22** (P=0.001)	0.12** (P=0.001)	0.14** (P=0.001)	0.091** (P=0.001)	0.6** (P=0.001)	0.32** (P=0.001)	0.22** (P=0.001)
Within Country	R ²	0.013	0.015	0.023	0.057	0.32	0.26	0.036
	Slope	-0.016** (P=0.0003)	-0.016** (P=0.0002)	-0.02*** (P=3.2e-06)	-0.033*** (P=2.2e-13)	-0.064*** (P=6e-06)	-0.08*** (P=5.6e-15)	0.058*** (P=3.40e-63)
	Mantels r	0.14** (P=0.001)	0.15** (P=0.001)	0.18** (P=0.001)	0.26** (P=0.001)	0.57** (P=0.001)	0.49** (P=0.001)	0.2** (P=0.001)
Within Region	R ²	0.038	0.0051	0.013	0.00072	0.25	0.074	0.013
	Slope	-0.044*** (P=2.5e-100)	-0.017*** (P=4.2e-15)	-0.028*** (P=2.1e-35)	-0.0072* (P=0.0021)	-0.089*** (P=4e-66)	-0.089*** (P=4e-66)	-0.019* (P=4.9e-3)
	Mantels r	0.2** (P=0.001)	0.079** (P=0.001)	0.12** (P=0.001)	0.034** (P=0.001)	0.51** (P=0.001)	0.27** (P=0.001)	0.0.12** (P=0.001)
Between Regions	R ²	0.00033	0.00041	0.001	0.0014	0.00044	0.035	0.00024

	Slope	0.006*** (P=4.2e-05)	0.0069*** (P=6.7e-06)	0.012*** (P=3.0e-14)	0.015*** (P=2e-16)	-0.01 (P=0.11)	-0.084*** (P=1.7e-102)	-0.0089* (P=3.5e-3)
	Mantels r	-0.02 (P=1)	-0.025 (P=1)	-0.042 (P=1)	-0.044 (P=1)	0.013 (P=0.22)	0.19** (P=0.001)	0.04** (P=0.001)

Table 4: Results of the linear regression models and statistical tests to investigate the association between resistome similarity and sampling distances. Mantel tests were performed between dissimilarity and distance matrices, and the distance-decay models used ln-transformed similarity matrices (1-dissimilarity). Dissimilarities were measured either by Aitchison distances for the Abundance models or Bray-Curtis distances for the assembly variant models. Asterisks: * $P \leq 0.01$, ** $P \leq 0.001$, and *** $P \leq 0.0001$.

Data input	Group	Comparison	Estimated Difference	Std. Error	t-ratio	P-value	
Abundance	Acquired	Within Country - Within Region	0.0282	0.0038	7.4643	2.71e-13***	
		Within country - Between Regions	-0.0216	0.0036	-6.0431	4.53e-09***	
		Within region - Between Regions	-0.04978	0.0024	-20.4194	0.0e+00***	
	FG	Within Country - Within Region	0.0039	0.0040	0.2520	9.66e-01	
		Within country - Between Regions	-0.0227	0.0038	-6.0289	4.95e-09***	
		Within region - Between Regions	-0.0237	0.0026	-9.2234	3.34e-14***	
	Other ARGs	Within Country - Within Region	0.00750	0.0042	1.8078	1.67e-01	
		Within country - Between Regions	-0.0325	0.0039	-8.2713	3.38e-14***	
		Within region - Between Regions	-0.040	0.0027	-14.9193	0.0e+00**	
	Bacteria	Within Country - Within Region	-0.0259	0.0045	-5.7449	2.76e-08***	
		Within country - Between Regions	-0.0476	0.0043	-11.1680	8.66e-15***	
		Within region - Between Regions	-0.0217	0.0029	-7.4605	2.78e-13***	
	Assembly variants	Acquired	Within Country - Within Region	0.0247	0.0178	1.3906	3.46e-01
			Within country - Between Regions	-0.0538	0.0175	-3.0711	6.08e-03*
			Within region - Between Regions	-0.0785	0.0089	-8.8021	3.90e-08***
FG		Within Country - Within Region	-0.0146	0.0107	-1.3631	3.60e-01	
		Within country - Between Regions	0.00390	0.0103	0.3777	9.24e-01	
		Within region - Between Regions	0.0185	0.0060	3.0965	5.57e-03*	

	Other ARGs	Within Country - Within Region	0.00395	0.0078	5.0274	1.48e-06***
		Within country - Between Regions	-0.0100	0.0074	-1.3544	3.65e-01
		Within region - Between Regions	-0.0495	0.0049	-10.1203	3.02e-14***

Table 5: Pairwise differences between the slopes of the linear regression models for distance-decays. The estimated difference represents the contrast between the slopes of two spatial scales, along with associated standard errors, test statistics (t-ratios), and p-values indicating the significance of each comparison. Asterisks: * $P \leq 0.01$, ** $P \leq 0.001$, and *** $P \leq 0.0001$.

Conclusion

Here, we present a brief overview of the content of the entire PanRes database and how we have distinguished between the different ARGs in our paper, “*Geographics and bacterial networks shape the global urban sewage resistome.*” We repeated the primary analyses on compositional data and the distance-decay analyses to demonstrate how the main conclusions of the paper have been influenced by our decision to focus only on specific groupings.

PanRes encompasses more than 14,000 genes that confer resistance to antimicrobials, biocides, and metals, representing a diverse array of genetic sequences. However, by combining so many different types of collections into one, we introduced a large uncertainty regarding gene annotations and biological origins. To study the latent reservoir of antimicrobial resistance in sewage resistomes, we focused on ARGs with well-curated discovery methodologies and phenotypic validation (Table 1). Specifically, we selected:

- ResFinder¹ for the mobilized, or acquired, ARGs and
- ResFinderFG² and the Daruka et al. (2025)³ for ARGs identified with functional metagenomics (FG), representing the latent reservoir.

Figures 1 and 2 showed that the FG ARGs had minimal overlap with the remaining PanRes ARGs, while the acquired ARGs appeared to be a subset of the broader collection ARGs.

Table 2 summarizes the read alignment counts across the three different groupings, where we observed that the other ARGs were the group with the most fragments aligned overall and per sample. However, the FG ARGs were still more abundant than the acquired ARGs. Similarly, we could see from the alpha diversity metrics that the FG ARGs showed higher richness and diversity than the acquired ARGs, where the other ARGs fell between the two (Figure 3). The abundance and beta diversity metrics showed similar behavior with some regional separation (Figure 4), which, taken together, show that the other ARGs may be more diverse, but their biological relevance for our study remains unclear.

The analyses of distance-decay patterns on both the abundance and the dispersal of variants detected in metagenomic assemblies showed that the other ARGs did not fundamentally alter our conclusions that the FG ARGs behave differently than the acquired ARGs in terms of geographic spread (Figures 5-6, Tables 4-5).

In conclusion, our analyses of the remaining other ARGs demonstrated that this group of genes consistently exhibited mixed characteristics across all of our analyses, suggesting that they represent a mixture of true mobilized, thus acquired, and non-mobilized ARGs. Furthermore, these genes lack a clear definition in contrast to our chosen subset, namely that they have not been laboratory-verified to cause phenotypic resistance. Therefore, we opted to exclude these genes from the main manuscript.

References

1. Bortolaia, V. *et al.* ResFinder 4.0 for predictions of phenotypes from genotypes. *Journal of Antimicrobial Chemotherapy* **75**, 3491–3500 (2020).
2. Gschwind, R. *et al.* ResFinderFG v2.0: a database of antibiotic resistance genes obtained by functional metagenomics. *Nucleic Acids Research* gkad384 (2023)
doi:10.1093/nar/gkad384.
3. Daruka, L. *et al.* ESKAPE pathogens rapidly develop resistance against antibiotics in vitro. *Nat Microbiol* 1–19 (2025) doi:10.1038/s41564-024-01891-8.
4. Martiny, H.-M. *et al.* ARGprofiler—a pipeline for large-scale analysis of antimicrobial resistance genes and their flanking regions in metagenomic datasets. *Bioinformatics* **40**, btae086 (2024).
5. Zankari, E. *et al.* Identification of acquired antimicrobial resistance genes. *Journal of Antimicrobial Chemotherapy* **67**, 2640–2644 (2012).
6. Alcock, B. P. *et al.* CARD 2023: expanded curation, support for machine learning, and resistome prediction at the Comprehensive Antibiotic Resistance Database. *Nucleic Acids Res* **51**, D690–D699 (2023).
7. Bonin, N. *et al.* MEGARes and AMR++, v3.0: an updated comprehensive database of antimicrobial resistance determinants and an improved software pipeline for classification using high-throughput sequencing. *Nucleic Acids Research* **51**, D744–D752 (2023).
8. Feldgarden, M. *et al.* AMRFinderPlus and the Reference Gene Catalog facilitate examination of the genomic links among antimicrobial resistance, stress response, and virulence. *Scientific Reports* **11**, (2021).
9. Gupta, S. K. *et al.* ARG-ANNOT, a New Bioinformatic Tool To Discover Antibiotic Resistance Genes in Bacterial Genomes. *Antimicrobial Agents and Chemotherapy* **58**, 212–220 (2014).
10. Pal, C., Bengtsson-Palme, J., Rensing, C., Kristiansson, E. & Larsson, D. G. J. BacMet: antibacterial biocide and metal resistance genes database. *Nucleic Acids*

Research **42**, D737–D743 (2014).

We would like to thank the reviewers for their time commenting on our study, and we have based on the response incorporated our analyses of the 'Other ARGs' into the supplementary material (line 157).

Reviewer #2 (Remarks to the Author):

I want to thank the authors for their thorough and constructive revisions. The additional analyses and clarifications have strengthened the manuscript and convincingly addressed the concerns raised in the previous review round.

Overall, this is an important contribution that will be of broad interest to the microbial genomics and public health communities. I am pleased with the revised version.

Reviewer #3 (Remarks to the Author):

In the third round of review, Martiny and colleagues have addressed the concerns me and other reviewers had by extending their analyses to the "Other ARGs" that weren't assigned to the categories in the main manuscript. This extra analysis is insightful, but currently only made available to the reviewers.

Thank you for so thoroughly addressing the reviewer remarks. I would suggest that these analyses are included in the supplementary information of the paper, so that they are also available to readers who may have the same questions as the reviewers did.

We have added the analyses of the other ARGs into the supplementary information for the manuscript and refer to this in line 157.

These are reviews for the paper by Martiny and colleagues entitled “Geographics and bacterial networks shape the global urban sewage resistome” (#NCOMMS-25-08182). The paper wields a large dataset to tussle with some huge important questions about the biogeography of antibiotic resistance genes and is remarkably resemblant of another distance-decay paper by an author of the same surname.

While the dataset is impressively large spanning >300 cities and the paper seems mostly methodologically sound, it is difficult to read, the distance-decay statistical models are incomplete, and presentation of the results are piecemeal or hurried.

The specific comments below are intended to help improve the manuscript.

I do like the overall structure of the results: summary of the metagenomes, geographic distribution, distance-decay, network analysis.

L80-81 Not clear on what a ‘large number’ is. About how many? Same for ‘considerably larger number’ on L85. Same for ‘large amount’ and ‘tiny fraction’ on L113.

The second paragraph of the introduction does a poor job of setting up the problem and the study.

The factors at play (regional [cultural, economic, etc.], distance, and bacterial (host) communities) are not disentangled clearly for the reader by the end of the paragraph.

The fourth paragraph could be trimmed of distracting points on the different databases to better lead to the hypothesis.

The hypothesis is also not quite clear because human-associated bacteria (L26) are also bacterial taxa (L25). Please clarify the terminology.

I assume the genes in ResFinder are assumed ‘acquired’ because they were most likely from genomes of isolated pathogens. And, because they are pathogenic, and I assume actively infecting a patient (maybe taking antibiotics) at the time they were isolated, they experienced selection pressure to ‘acquire’ antimicrobial resistance. This is different from metagenomics which targets mostly non-pathogenic bacteria. Therefore the selection pressure to acquire antimicrobial resistance is much lower. More likely is that these ARGs play a non-pathogenic ‘functional’ role, possibly even not related to antimicrobials at all. Therefore, these genes are more controlled by environmental conditions suitable for growth of the host (pH, temperature, oxygen, osmolarity, nutrient availability, etc.) akin to Baas Becking’s hypothesis. It follows that human associated (opportunistic) pathogens might disperse more widely by hitching rides with humans. I don’t want to rewrite the introduction, but the authors have to straighten this out.

L133-134 This seems to ‘confirm’ the opposite of the hypothesis already. Please clarify.

L134 Remove ‘highly’.

L135 You'll want to make this clear that this is relative abundance (vs. absolute abundance)

L152 "spun down" is informal. Also, what g force and for how long?

L308-309 'named' vs 'unnamed' may be a little vague. Maybe 'latinized' vs 'placeholder' would be more specific and has been used in one other case I know of:
<https://gtdb.ecogenomic.org/stats/r220>

L340 Did the PCA and permanova explain the same amount of variation? What was the permanova's R-squared value?

L343 Procrustes analysis compares two arrangements in *reduced space*, where the arrangements are always imperfect representations of the calculated dissimilarities. Wouldn't it be better to test the differences of the dissimilarities directly with something like a mantel test?

L362 Many decay rates are referred to, but only one is reported. What test are the p-values from? The stats in this section need some work. Could a more sophisticated model, just adding an interaction term (within or between different countries), maybe, be more direct in asking about differences between slopes? If not, maybe confidence intervals (with `confint()`) maybe) can be used to gauge differences in slope? More description in the Table S2 caption would also really help.

Table S2 Some symbols are not reproducing in the table leading to ambiguity in what the authors mean. Mantel tests are usually done with two dissimilarity matrices. Which two are being compared in each row? Plus, these aren't summary stats. They are results of hypothesis testing.

Table S3 More description of these data are necessary.

L369 It may have affected it, but more specifically, you didn't *detect* any effect.

Fig2abc & Fig3cd It is difficult to see the pink/orange circles compared to the other points. For completeness, it would be great to have in the caption or somewhere locally the p-values for the regression lines, but also to know if the slopes significantly differed.

Fig 4 These figures are very busy and could benefit from some reduction and/or some visual clarification (e.g., grouping). The legend is too small in Fig4a to make much use of.

Table S1 The R-squared values are all pretty low, despite significance.

L362,364,369,533 From what I can tell, even after reviewing the code, differences in slope were never directly tested by including an interaction term, so these statements aren't justified. What is function `fit_dd_model2()` do?